# Conditional simulation of spatial rainfall fields using random mixing: A study that implements full control over the stochastic process

Jieru Yan[1,*], Fei Li[1,*], András Bárdossy[2], and Tao Tao[1]

[1]College of Environmental Science and Engineering, Tongji University, Shanghai, China.
[2]Institute of Modeling Hydraulic and Environmental Systems, Department of Hydrology and Geohydrology, University of Stuttgart, Stuttgart, Germany.
[*]These authors contributed equally to this work.

**Correspondence:** Tao Tao (taotao@tongji.edu.cn)

**Abstract.** The accuracy of spatial precipitation estimates with relatively high spatiotemporal resolution is of vital importance in various fields of research and practice. Yet the intricate variability and the intermittent nature of precipitation make it very difficult to obtain accurate spatial precipitation estimates. Radar and rain gauge are two complementary sources of precipitation information: the former is inaccurate in general but is a valid indicator for the spatial pattern of the rainfall field; the latter is relatively accurate but lacks spatial coverage. Several radar-gauge merging techniques have been proposed in the scientific literature to obtain spatial precipitation estimates. Conditional simulation has great potential to be used in spatial precipitation estimation. Unlike the commonly used interpolation methods, the results from conditional simulation are a range of possible estimates due to its Monte Carlo framework. Yet an obstacle that hampers the application of conditional simulation in spatial precipitation estimation is obtaining the marginal distribution function of the rainfall field with sufficient accuracy. In this work, we propose a method to obtain the marginal distribution function from radar and rain gauge data. The conditional simulation method, random mixing (RM), is used to simulate rainfall fields. The radar and rain gauge data used for the application of the proposed method are derived from a stack of synthetic rainfall fields. Due to the full control over the stochastic process, the accuracy of the estimates is verified comprehensively. The results from the proposed approach are compared with those from 3 well-known radar-gauge merging techniques: ordinary Kriging, Kriging with external drift, and conditional merging, and the sensitivity of the approach towards the factors - number of rain gauges and random error in radar estimates - is analyzed in the same experimental context.

## 1 Introduction

Precipitation is one of the most important factors in hydrology and meteorology. The accuracy of spatial precipitation estimates with relatively high spatiotemporal resolution is of vital importance in various fields of research and practice, such as the

promotion of meteorological and hydrological monitoring, forecast to enhance the capability to cope with natural disasters, the study of climate trends and variability, and the management of water resources (Yilmaz et al., 2005; Michaelides et al., 2009; Jiang et al., 2012; Liu et al., 2017). Yet unlike many other hydro-meteorological variables such as temperature and humidity, precipitation is intermittent in space and time, i.e., non-rainy areas amidst rainy areas and dry periods amidst wet periods

continue to exist (Kumar and Foufoula-Georgiou, 1994). The intricate spatiotemporal variability and the intermittent nature of precipitation make it very difficult to obtain accurate spatial precipitation estimates (Emmanuel et al., 2012; Cristiano et al., 2017).

Rain gauges, as the only direct measurement device of precipitation on the ground surface, are still counted as the most reliable source of precipitation information in hydrology. Yet rain gauges are only available at limited locations. The repre-

sentativeness of gauge observations for the entire precipitation field is therefore limited. Significant research has shown that precipitation estimation based on gauge observations suffers from degraded levels of accuracy with increased rainfall intensities during storms where convective processes are significant (Adams, 2016). Meteorological radar, on the other hand, is a superb tool for measuring spatial patterns of reflectivity at the altitude where the measurement is taken. Yet radar-based precipitation estimation can be problematic due to various sources of errors, e.g., variations in the vertical profile reflectivity (VPR),

static/dynamic clutter, signal attenuation, anomalous propagation of the beam, uncertainty in the Z–R relationship, etc. (see Doviak and Zrnić, 1993; Collier, 1999; Fabry, 2015, for details). Despite the various sources of errors, radar-based precipitation estimation has been appreciated as a valid indicator of precipitation patterns (Méndez Antonio et al., 2009; Yan et al., 2020). In summary, radar and rain gauge are two complementary sources of precipitation information: the former is in general inaccurate but a valid indicator for the spatial pattern of the rainfall field; the latter is relatively accurate but lacks spatial coverage.

Considering the pros and cons of the two sources of precipitation information, many radar-gauge merging techniques have been developed to obtain spatial precipitation estimates over the past years. Wang et al. (2013) grouped these techniques into two categories: bias reduction techniques and error variance minimization techniques. The bias reduction techniques attempt to correct the bias of radar rainfall estimates using rain gauge observations. This class of techniques has a long history, from the earliest mean field bias correction schemes where a single correction factor is applied to the entire radar field (e.g., Wilson,

1970), to the later local bias correction schemes where spatially distributed correction factors are applied (e.g., Brandes and Edward, 1975; James et al., 1993; Michelson and Koistinen, 2000). The latter, the error variance minimization techniques attempt to eliminate the bias of radar rainfall estimates while minimizing the variance between radar and rain gauge measurements. Representatives of this class include the Bayesian data combination scheme (Todini, 2001), the Kriging-based techniques such as conditional merging (Sinclair and Pegram, 2005), Kriging with external drift (Hengl et al., 2003; Haberlandt, 2007;

Velasco-Forero et al., 2009), and co-Kriging (Schuurmans et al., 2007; Sideris et al., 2014).

In addition to the two categories mentioned above, we pay particular attention to a class of methods that simulate spatial random fields under the Monte Carlo framework. The logic behind is straightforward. When faced with uncertainty in the process of making a forecast or estimation, rather than replacing the uncertain variable with a single average number, the simulation might prove to be a better solution by providing a range of possible outcomes. In the context of merging radar and rain

gauge data, the simulation is often applied under constraints, such as the equality constraints at rain gauge locations, the field

pattern indicated by radar, etc. There are several methods that simulate spatial random fields with given covariance functions in Gaussian space, such as turning bands simulation, LU-decomposition-based methods, sequential Gaussian simulation, etc. (Mantoglou and Wilson, 1982; Deutsch and Journel, 1998; Chilès and Delfiner, 2000; Lantuéjoul, 2002). Yet studies on conditional simulation of rainfall fields are rare. One of the major obstacles that hampers the application of conditional simulation in spatial precipitation estimation is how to obtain the marginal distribution function of the rainfall field with sufficient accuracy. The distribution function is essential to transform the simulated Gaussian fields to rainfall fields of interest as the simulation is normally embedded in Gaussian space. Given this, we propose a method to obtain the distribution function from radar and rain gauge data.

In this paper, the method employed to simulate rainfall fields is random mixing (RM), which was first proposed by Bárdossy and Hörning (2016) to solve inverse modeling problems in the modeling of groundwater flow and transport. RM inherits the concept of the gradual deformation (GD) approach described in (Hu, 2000) that a conditional field of interest is obtained as a linear combination of unconditional random fields. Yet different from GD, RM is targeted and flexible to incorporate different kinds of constraints - linear or nonlinear - and the utilization of spatial copulas in the description of the underlying dependence structure enables the separate treatment of dependence structure and marginal distribution.

The radar and rain gauge data used for the application of the proposed approach are derived from a stack of synthetic rainfall fields. Unlike the commonly used verification method, e.g., leave-n-out cross-validation where one verifies the accuracy of the estimates at limited locations, in this study the accuracy of the estimates is verified more comprehensively due to the full control over the stochastic process. The results from the proposed approach are compared with those from several well-known radar-gauge merging techniques: ordinary Kriging, Kriging with external drift, and conditional merging. Finally, the sensitivity of the proposed approach towards the factors - number of rain gauges and random error in the radar estimates - is analyzed.

This paper is divided into 6 parts. After the introduction, the methodology of RM is elaborated in Sect. 2. Sect. 3 describes the data used in this study. Sect. 4 compares the results from the proposed approach with those from other techniques and analyzes the sensitivity of the approach. Sect. 5 describes the scope and assumption of the approach and discusses the limitation of this study. Sect. 6 ends this paper with conclusions and outlook.

## 2 Methodology

### 2.1 CDF of the rainfall field

As has been described in the introduction, one of the major obstacles that hampers the application of conditional simulation in spatial precipitation estimation is obtaining the cumulative distribution function of the rainfall field (abbr. rainfall CDF hereafter) with sufficient accuracy. The rainfall CDF is essential to transform the simulated Gaussian fields to rainfall fields of interest. In this subsection, an algorithm to compute the rainfall CDF from radar and rain gauge data is proposed, which is specified as follows:

1. Estimate the spatial intermittency of the rainfall field as the ratio of the number of dry pixels to the total number of pixels in the radar estimates $R_r$. We denote the estimated spatial intermittency as $u_0$.

2. Transform the radar estimates $R_r$ to non-exceedance probabilities (denoted as quantiles hereafter), which results in a quantile map. Due to the intermittent nature of precipitation, all the dry pixels in $R_r$ are transformed to $u_0$, i.e., $u_0$ is the smallest value in the quantile map.

3. Determine the gauge-respective quantiles in the radar quantile map, denoted as $u_k$ for $k = 1, \ldots, K$. The two datasets - rain gauge observations $r_k$ and gauge-respective quantiles $u_k$ - form $K$ pairs $(r_k, u_k)$. Take the following quality control steps for these pairs:

   (1) Check the consistency at zeros and remove those pairs whenever a zero is encountered, i.e., $r_k = 0$ or $u_k = u_0$. In the ideal case when the radar estimates can perfectly represent the pattern of the rainfall field, zero gauge observations and the smallest quantile $u_0$ should co-exist. Yet in practice, there exist various factors that can reduce the representativeness of the radar-indicated field pattern. A zero gauge observation could be collocated with a quantile that is slightly larger than $u_0$, and a dry pixel could be collocated with a gauge observation that is slightly larger than 0 mm. The consistency at zeros is an important indicator of the mismatch between radar and rain gauge data, though the zeros in both datasets are not used in the computation of the rainfall CDF.

   (2) Maintain the consistency in terms of the order. In the ideal case as described in (1) the pairs $(r_k, u_k)$ represent $K$ points being exactly on the rainfall CDF. Yet due to the degraded representativeness of the radar estimates, the Spearman's rank correlation of $r_k$ and $u_k$ can be less than 1. Namely, the two datasets have different orders, for example, the largest rain gauge observation does not correspond to the largest radar quantile. To maintain a consistent order, the remnant values in $r_k$ and $u_k$ (after applying (1)) are sorted in ascending order.

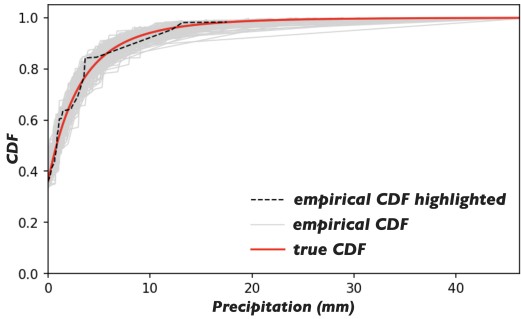

**Figure 1.** Light grey: The empirical CDF(s) obtained by linearly joining $(r_k, u_k)$ after the two quality control steps, with one CDF highlighted by the black dashed line. Red: the true rainfall CDF. Note that the empirical CDF(s) shown here are computed from the synthetic data as described in Sect. 3.

Note that both consistencies - consistency at zeros and consistency in terms of the order - are good indicators of the mismatch between radar and rain gauge data. A significant mismatch, e.g., the collocation of a dry pixel with a 5 mm rainfall record, or a low Spearman's rank correlation (say $\rho_r < 0.8$), can lead to unreliable estimates.

4. After the quality control steps, it is assumed that the set of points $(r_k, u_k)$ are distributed without bias around the true CDF, see the empirical CDF obtained by linearly joining the points in Fig. 1. The rainfall CDF is obtained by fitting a theoretical CDF model under the condition that the fitted CDF starts to be positive at the point $(0, u_0)$. The estimated rainfall CDF is denoted as $G(\cdot)$ hereafter.

In the above algorithm, the radar data provides a hint on the representativeness of the rain gauge data. For example, has the extreme of the rainfall field been properly sampled by the gauges? If not, to what extent has the extreme been underestimated by the samples (rain gauge observations)? One could answer the question by checking the maximum value in the gauge-respective radar quantiles. Similarly, one could also find the answers to questions such as whether the samples are uniformly distributed in terms of the quantile range or just gather around the lower/higher range of the rainfall field. Without the additional information provided by radar, one would probably assign evenly distributed quantiles to the rain gauge observations as one usually does in the acquisition of the empirical CDF.

## 2.2 Prepare the constraints

The simulation is embedded in Gaussian space; hence the constraints should be transformed to the standard normal marginal (normalized Gaussian). Specifically, we consider the following 3 constraints:

(1) The equality constraints at rain gauge locations, when defined in terms of the standard normal marginal, mean the simulated values at the rain gauge locations should equal the values mapped from the rain gauge observations $r_k$:

$$Z(x_k) = \Phi^{-1}(G(r_k)) \quad \text{for } k = 1, \ldots, K. \tag{1}$$

where $Z$ is the simulated Gaussian field; $x_k$ denotes the rain gauge location; $G(\cdot)$ and $\Phi(\cdot)$ denote the rainfall CDF and the CDF of the standard normal distribution, respectively.

(2) The simulated Gaussian field $Z$ should preserve a given correlation function, which is obtained from the transformed radar estimates

$$Z'_r = \Phi^{-1}(G(R_r)) \tag{2}$$

The problem arises here that from the radar estimates $R_r$, one can only obtain a truncated Gaussian field $Z'_r$, as all the dry pixels in $R_r$ are converted to $z_0 = \Phi^{-1}(u_0)$ where $u_0$ is the spatial intermittency. The sill of the correlation function evaluated from $Z'_r$ is reduced due to the truncation. Fig. 2 displays the empirical variograms evaluated from 1000 truncated Gaussian fields compared with the true variogram used to generate the corresponding continuous Gaussian fields. From the figure, it can be seen that the empirical and the true variograms have very similar patterns. Practically,

the true correlation function can be approached by scaling given a priori knowledge that the variance of the simulated field is 1. Besides, as random mixing is a geostatistical simulation method, the choice of the correlation function has a limited effect on the estimates just as the case in Kriging (Verworn and Haberlandt, 2011). We denote the estimated correlation function as $\Gamma$ hereafter.

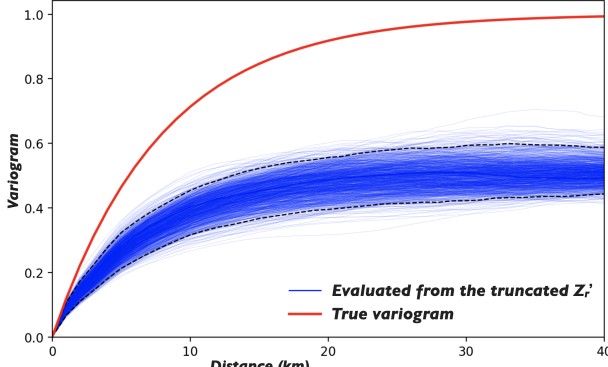

**Figure 2.** The blue lines denote the empirical variograms evaluated from 1000 truncated Gaussian fields, where the 95% confidence interval is marked by the black dashed lines. The red line denotes the true variogram used to generate the continuous Gaussian fields. Note that the continuous Gaussian field is truncated at $z_0 = \Phi^{-1}(0.36)$.

Note that "correlation function" and "variogram" are used interchangeably here, as it is common in geostatistics to work with variogram, whose estimation is shown to be more stable than that of the correlation function (Calder and Cressie, 2009). Namely, one simulates by using the correlation function as the measure of spatial dependence, yet the spatial dependence of the simulated field is normally examined on its variogram.

(3) The pattern of the simulated field should resemble that of the radar estimates as closely as possible. This forms an optimization problem, and the goal is to maximize the Pearson's correlation coefficient of the simulated field $Z$ and the reference field $Z'_r$, i.e.,

$$\mathcal{O}(Z) = \rho(Z, Z'_r) \to \max \tag{3}$$

where $Z$ is the simulated Gaussian field, $Z'_r$ is obtained from the radar estimates as defined in Eqn (2), and $\rho(\cdot)$ is the Pearson's correlation coefficient. The same problem arises: $Z$ is continuous, while $Z'_r$ is truncated. To evaluate the objective function, surely one could truncate $Z$ with $z_0$. Yet one could also use $Z$ directly as shown in Eqn (3). The difference is minor, as a high correlation between $Z'_r$ and $Z$ means a high correlation between $Z'_r$ and $Z$-truncated. From a parsimonious point of view, we use $Z$ directly in the evaluation of the objective function.

## 2.3 Random mixing

The task is to estimate the true rainfall field given the set of rain gauge observations and the radar estimates. In terms of conditional simulation, this means to simulate a Gaussian field $Z$ that fulfills all the constraints described in Sect. 2.2, and then convert $Z$ to the rainfall field of interest, i.e., to obtain an estimate of the true rainfall field.

We use random mixing (RM) to fulfill the task. RM utilizes the concept described in Hu (2000) that the conditional field of interest is obtained as a linear combination of many unconditional random fields:

$$Z = \sum_{i=1}^{N} \alpha_i Y_i \tag{4}$$

where $Y_i$(s) are independent Gaussian random fields with identical statistical properties, i.e., the marginal distributions all follow the standard normal distribution and with the same correlation function. If in addition, the L2 norm of the weights $\alpha_i$ fulfills

$$\sum_{i=1}^{N} \alpha_i^2 = 1 \tag{5}$$

then the resultant conditional field $Z$ is statistically identical to $Y_i$, see Bárdossy and Hörning (2016) for the demonstration. There are several methods to obtain Gaussian random fields with the given correlation function. We have used an efficient method - the fast Fourier transformation for regular grids (Wood and Chan, 1994; Wood, 1995; Ravalec et al., 2000).

It is necessary to differentiate Hu's method from the proposed one before specifying the algorithm. Hu's method (Hu, 2000; Hu et al., 2001) incorporates linear data using conditional Kriging, and the method is extended to combine dependent conditional fields in Hu (2002), while RM incorporates linear or nonlinear constraints under the unified concept of randomly mixing unconditional random fields. The algorithm of RM is specified as follows:

1. The conditional Gaussian field $Z$ that has a prospect to fulfill all the constraints is obtained as

$$Z = \sum_{i=1}^{N} \alpha_i Y_i + (\cos\theta \cdot H + \sin\theta \cdot H') \cdot \sqrt{1 - \sum \alpha_i^2} \tag{6}$$

$Z$ is constructed by two parts:

   (a) The first part

$$\sum_{i=1}^{N} \alpha_i Y_i$$

is made up of $N$ statistically identical unconditional random fields $Y_i$(s) with the correlation function $\Gamma$, as estimated in Sect. 2.2. The role of this part is to fulfill the equality constraints at the rain gauge locations. Thus $K$ linear constraints are imposed as

$$\sum_{i=1}^{N} \alpha_i \cdot Y_i(x_k) = \Phi^{-1}(G(r_k)) \quad \text{for } k = 1, \ldots, K. \tag{7}$$

See Eqn. (1) for the definitions of $x_k$, $r_k$, $G(\cdot)$ and $\Phi^{-1}(\cdot)$. In total, we have $N$ unknowns: $\alpha_i$ for $i = 1, \ldots, N$, and $K$ equations. If $N > K$, this forms an under-determined equation system. Multiple techniques exist to solve such a system. Specifically, we found the set of weights with the least L2 norm, i.e., minimized $\sum_{i=1}^{N} \alpha_i^2$ by using the singular value decomposition technique.

The constraint $\sum_{i=1}^{N} \alpha_i^2 < 1$ is imposed in addition to ensure that the second part has a positive weight, i.e., $1 - \sum \alpha_i^2 > 0$. The extra constraint is satisfied further by increasing $N$, i.e., increasing the degree of freedom for the equation system.

(b) The second part

$$(\cos\theta \cdot H + \sin\theta \cdot H') \cdot \sqrt{1 - \sum \alpha_i^2}$$

is made up of two independent, statistically identical conditional random fields $H$ and $H'$ which are referred to as $H$-field hereafter. The $H$-field is also obtained as a linear combination of unconditional random fields $Y_i'$ (statistically identical to $Y_i$):

$$H = \sum_{i=1}^{M} \beta_i Y_i'.$$

The $H$-field is special because of the zeros at the rain gauge locations, such that the addition of the second part to the first part does not change the values at the rain gauge locations; hence the equalities in Eqn. (7) can be re-written as

$$Z(x_k) \equiv \sum_{i=1}^{N} \alpha_i \cdot Y_i(x_k) + 0 = \Phi^{-1}(G(r_k)) \quad \text{for } k = 1, \ldots, K. \tag{8}$$

The remaining question is how to obtain such $H$-fields. Similarly, an under-determined system is created as

$$\sum_{i=1}^{M} \beta_i \cdot Y_i'(x_k) = 0 \quad \text{for } k = 1, \ldots, K \tag{9}$$

with $M$ unknowns, $K$ equations, and $M > K$. To ensure that the $H$-field is statistically identical to $Y'$ (or $Y$), the following constraint is imposed additionally:

$$\sum_{i=1}^{M} \beta_i^2 = 1 \tag{10}$$

The set of weights ($\beta_i$ for $i = 1, \ldots, M$) can be determined by solving Eqns. (9) first, and then scaling the weights with the factor $(1/\sqrt{\sum \beta_i^2})$ such that Eqn. (10) is satisfied further. Because of the 'zeros', scaling does not change the values at the rain gauge locations.

The Gaussian field $Z$ obtained from Eqn. (6) fulfills the equality constraints at the rain gauge locations and reproduces the correlation function $\Gamma$. The correlation function is reproduced because the L2 norm of the weights of all the fields underlying $Z$ satisfies

$$\sum \alpha_i^2 + (\cos^2\theta + \sin^2\theta) \cdot (1 - \sum \alpha_i^2) = 1. \tag{11}$$

It should be noted that one $H$-field is already enough to do the same job. For example, one could obtain such a $Z$ by

$$Z = \sum_{i=1}^{N} \alpha_i Y_i + H \cdot \sqrt{1 - \sum \alpha_i^2}$$

The aim of using two $H$-fields instead of one is to introduce more freedom such that the third constraint can be met additionally. The relative weights of the two parts ($\sqrt{\sum \alpha_i^2} : \sqrt{1 - \sum \alpha_i^2}$) do matter. It is desired that the second part weighs more, such that it is favorable for the solution of the optimization problem defined in the next step. Thus when solving the under-determined system in (a), the set of weights $\sqrt{\sum \alpha_i^2} \ll 1$ is found.

2. Due to the freedom introduced by the extra $H$-field, the Gaussian field $Z$ obtained from Eqn. (6) is a function of $\theta$, and so does the objective function evaluated from $Z$, which actually defines an 1D optimization problem w.r.t. $\theta$:

$$\mathcal{O}(\theta) = \rho\big(Z(\theta), Z'_r\big) \to \max \tag{12}$$

with $\theta \in (-\pi, \pi]$. See the definitions of $Z'_r$ and $\rho(\cdot)$ in Eqns. (2) and (3). The task is to find the $\theta$ that produces the maximum objective function value. Various methods exist to solve the 1D unconstrained optimization problem. In this work, we simply used the trial-and-error method, and a coarse search followed by a fine search is what has been implemented to accelerate the solving process. The solution to the 1D optimization problem is denoted as $\theta^*$.

3. If $Z(\theta^*)$ meets the stopping criterion of the optimization, continue with Step 4. Otherwise, update $H$ and $H'$ as follows, and go back to Step 2.

$$H \leftarrow \cos\theta \cdot H + \sin\theta \cdot H' \tag{13}$$

$$H' \leftarrow \text{A newly generated } H\text{-field} \tag{14}$$

As always in optimization, there are multiple choices of the stopping criteria, such as the number of iterations, the pre-set limit of the objective function, the rate of decrease of the objective function, and so forth. Specifically, we have adopted the number of continuous iterations without improvement as the stopping criterion.

4. Finally, an estimate of the true rainfall field is obtained as

$$R = G^{-1}\big(\Phi(Z(\theta^*))\big) \tag{15}$$

## 3 Data

An artificial experiment was carried out to test the capability of the proposed approach at estimating the true rainfall field. Due to the lack of knowledge on the true fields, we have used synthetic ones: 1000 rainfall fields were generated independently and served as the 'true' rainfall fields, from which radar and rain gauge data were derived.

The motivation for doing so is that one can hardly verify the accuracy of the estimates comprehensively without the knowledge of the true rainfall field. Some studies employ the leave-n-out cross-validation method for verification where one verifies

the accuracy of the estimates at limited points. Yet compared to the accuracy at limited points, the accuracy of the estimates in terms of the overall statistics, e.g., the mean and maximum of the rainfall field, is more hydrologically interesting.

## 3.1 Generate the 'true' rainfall fields

1000 rainfall fields with the grid size 80×80 were generated. Each pixel is assumed to be representative of a 1×1 km$^2$ area, and all the 1000 fields were generated independently with identical properties, i.e., identical spatial intermittency, rainfall CDF, and correlation function. The generation procedure is specified as follows:

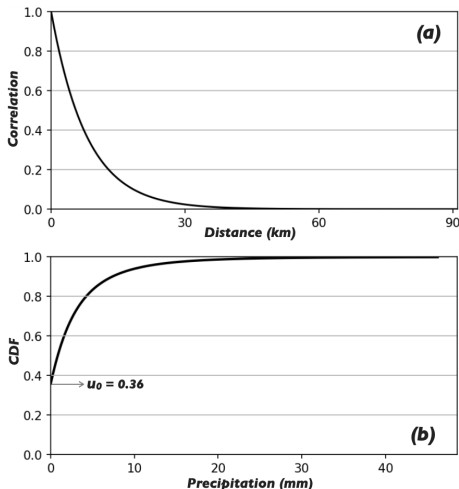

**Figure 3.** (a) The exponential correlation function used in the generation of the Gaussian random fields $Z_T$. (b) The rainfall CDF used in the generation of the true rainfall fields, where the spatial intermittency $u_0$ is labeled.

1. Generate 1000 Gaussian random fields $Z_T$ with a given correlation function. Fig. 3 (a) displays the exponential correlation function used in the generation of $Z_T$. Note that the subscript "$_T$" is used throughout this paper to denote the true Gaussian/rainfall fields, or further the true rainfall CDF. Similarly, we have used the fast Fourier transformation for regular grids to generate $Z_T$.

2. Generate a rainfall CDF where the lognormal distribution has been used as the model of the rainfall CDF, as this distribution is shown to be effective in describing the marginal distribution of rainfall rates or short-time rainfall (with 10 or 15 minutes' accumulation time) over an area (Bell, 1987; Crane and Robert, 1990; Pegram et al., 2001). Fig. 3 (b) shows the rainfall CDF used in the generation of the 1000 true rainfall fields, i.e., the true rainfall CDF: $G_T^{-1}(\cdot)$.

3. Convert the Gaussian random fields $Z_T$ to rainfall fields using the normal-score transformation method:

$$R_T = G_T^{-1}\big(\Phi(Z_T)\big) \tag{16}$$

where $R_T$ is the true rainfall field and $\Phi(\cdot)$ is the CDF of the standard normal distribution. Note that a quantile value $u_0$ is used to maintain the spatial intermittency, as labeled in Fig. 3 (b). Hence, all the pixel values that are smaller than $z_0 = \Phi^{-1}(u_0)$ in $Z_T$ are converted to zero (precipitation) in $R_T$.

## 3.2 Generate the radar estimates

The radar estimates were derived from the 'true' rainfall fields. Specifically, the Gaussian field $Z_T$ generated in Sect.3.1 was used. To mimic two commonly seen errors in radar estimates - random and nonlinear errors - the following procedure was applied:

1) Introduce a random error. The proposed approach assumes that the radar estimates measured aloft can represent the pattern of the rainfall field on the ground. Yet some factors do exist that can reduce the representativeness of the radar estimates, such as evaporation, complex terrain effects, and anthropogenic effects. A random error is therefore introduced to mimic this kind of error. We also utilize the concept of random mixing that the Gaussian field with the random error introduced is obtained by mixing two fields:

$$Z_r = w_1 \cdot Z_T + w_2 \cdot Z_e \tag{17}$$

where $Z_T$ is the 'true' Gaussian field and $Z_e$ is an independently generated Gaussian random field with the same statistical properties as $Z_T$. The constraint $w_1^2 + w_2^2 = 1$ is imposed in addition to ensure that the resultant $Z_r$ is statistically identical to $Z_T$ (or $Z_e$). The ratio of the two weights $(w_1/w_2)$ is used to measure the significance of the random error in analogy with a commonly seen parameter Signal-Noise Ratio (SNR). The proposed approach was tested on three levels of significance: SNR = 3, 5, and 10. Accordingly, the random error-introduced Gaussian fields were obtained as:

$$Z_r = 0.9487 \cdot Z_T + 0.3162 \cdot Z_e$$

$$Z_r = 0.9806 \cdot Z_T + 0.1961 \cdot Z_e$$

$$Z_r = 0.9850 \cdot Z_T + 0.0985 \cdot Z_e$$

2) Convert the random error-introduced Gaussian fields to rainfall fields using the normal-score transformation:

$$R_r = G_T^{-1}(\Phi(Z_r)) \tag{18}$$

The resultant $R_r$ differs from the true rainfall field in terms of the pattern, while the differences in terms of the statistical properties are small. See the statistical properties evaluated from the 'true' and the random error-introduced rainfall fields in Fig. 4.

3) Apply a nonlinear transformation to mimic the error induced due to the employment of an erroneous Z-R relationship. In practice, the accurate Z-R relationship is difficult to achieve. The omnipresent Z-R relationship $Z = 200R^{1.6}$ given by Marshall and Palmer (1948) is widely used in radar hydrology to convert radar reflectivity to rain rate. Long lists

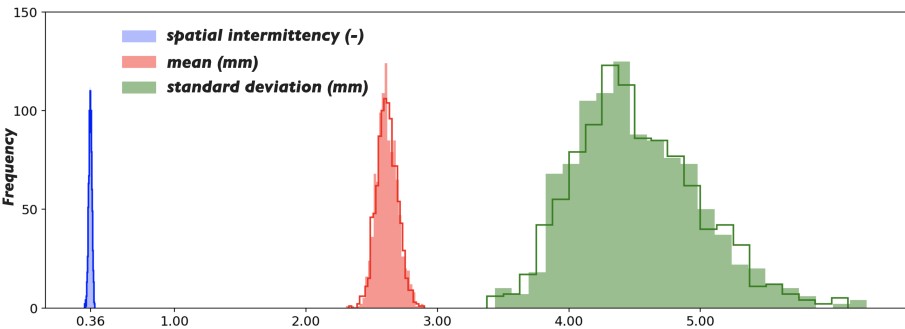

**Figure 4.** The color-filled histograms are evaluated from the 1000 'true' rainfall fields. The unfilled histograms on top of the color-filled ones are valuated from the 1000 random error-introduced rainfall fields (SNR = 3).

of vastly different Z-R relationships have been derived for different areas in different conditions in scientific literature (Uijlenhoet, 2001; Fabry, 2015). Yet the Z-R relationship is changing in space and time. Generally, there is no means to achieve the accurate one in real time. In other words, most of the time an erroneous Z-R relationship is employed, which results in a nonlinear departure of the radar estimates from the true rainfall field: the larger the measurement from radar, the more serious the departure. Given the above, we have used the following power function to mimic this nonlinear departure (where the operator '←' means updating):

$$R_r \leftarrow 0.87\, R_r^{0.83} \tag{19}$$

The choice of the two parameters - factor 0.87 and exponent 0.83 - is indeed arbitrary, as it makes no difference for the proposed approach where the transformed radar estimates - radar quantiles - are utilized. The monotonic transformation above does not change the quantile map. We have modeled a case when radar underestimates the precipitation, because radar data are prone to underestimate the precipitation (Curry, 2012; Berne and Krajewski, 2013; Shehu and Haberlandt, 2020), and underestimated precipitation is useless and can have negative effects for many hydrological applications. Yet the choice of the two parameters matters for radar-gauge merging techniques where the radar estimates are used directly. An underestimation in the radar estimates leads to an underestimate, for example.

### 3.3 Generate the rain gauge observations

The rain gauge observations were sampled from the true rainfall field $R_T$ at the 'rain gauge' locations. Due to the local effect of the rain gauge observations: the closer an ungauged location from the nearby rain gauge, the less uncertain the corresponding estimate is, it is favorable to have a denser rain gauge network. Yet it is always an interesting question to ask how dense should the rain gauge network be to achieve a sufficient accuracy. We try to answer this question in the experimental context of this study. As it would be too intricate to model the varied real-world rain gauge networks, which are usually irregularly distributed with various densities, we have made things as simple as possible. The proposed approach was tested on 3 layouts: 5×5, 6×6,

and 7×7 rain gauges that are uniformly distributed in the domain of interest (abbr. G25, G36, and G49 hereafter). Presumably, this gives an approximate coverage of one rain gauge for every 256, 178, and 131 km$^2$, respectively.

## 4 Results

### 4.1 Results Comparison

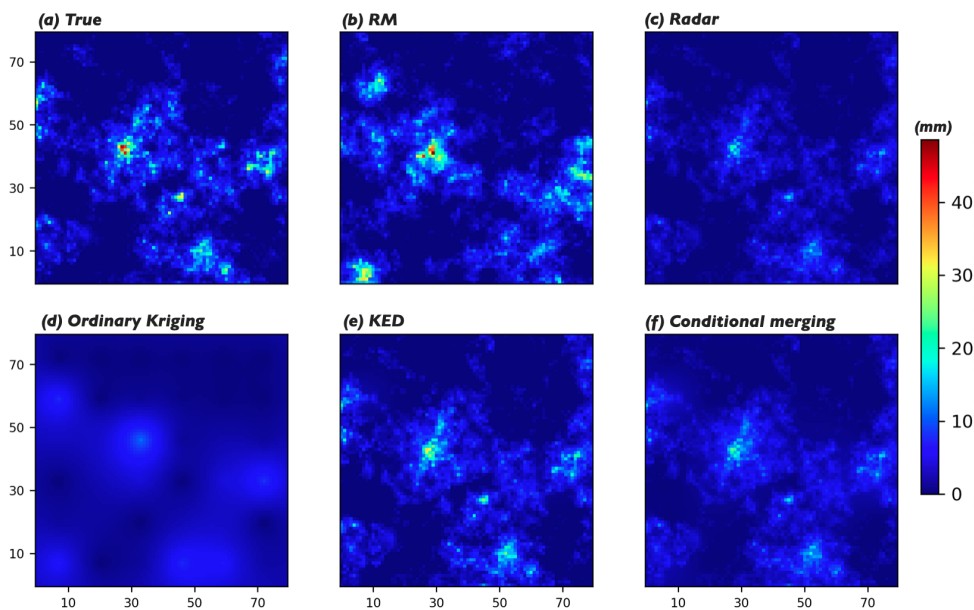

**Figure 5.** (a) The true rainfall field; (b) a single realization from RM (Scenario: G36, SNR=3); (c) the radar estimates; (d), (e), and (f) the estimates obtained from ordinary Kriging, Kriging with external drift (KED), and conditional merging, respectively.

The proposed approach (referred to simply as RM in this section, though in fact RM is only part of the approach) was used to estimate the 1000 true rainfall fields based on the corresponding radar estimates and rain gauge observations. The results from RM are compared with those from 3 well-known Kriging methods: ordinary Kriging, Kriging with external drift (KED), and conditional merging (Sinclair and Pegram, 2005). Surely it is not possible to display all the results. We pay attention to the results for one rainfall field which is randomly drawn from a total of 1000. In Fig. 5, a single realization obtained from
RM is shown in Panel (b), in comparison with the corresponding true rainfall field and radar estimates in Panels (a) and (c), respectively. Meanwhile, the estimates obtained from ordinary Kriging, KED, and conditional merging are displayed in (d), (e), and (f), respectively. The results shown here are typical enough to draw the following conclusions (Conclusion 1 for ordinary Kriging, 2 for KED and conditional merging, and 3 for RM):

1. The rainfall field obtained from ordinary Kriging misses both the pattern and the extremes of the true rainfall field, as
the method only considers the rain gauge observations, while the radar estimates do not contribute to the final estimates.

2. The estimates from KED and conditional merging capture the pattern, yet they both miss the extremes of the true field. KED outperforms conditional merging in this case, as KED takes the radar estimates as the external drift and tries to capture the linear relationship between the gauge observations and the radar estimates at the gauge locations. Thus the estimates from KED correct the extremes of the radar estimates a little bit, but not perfectly, as the scenario is that radar underestimates the rainfall field nonlinearly. In this specific case, the maximum values in the true and KED-estimated rainfall fields are 46.24 mm and 32.55 mm, respectively. Compared to KED, conditional merging is more dependent on the accuracy of the radar estimates, as the method assumes that the radar data provides an estimate of the actual Kriging error. As KED outperforms ordinary Kriging and conditional merging in this context, the results from RM are compared with those from KED in the following subsections.

3. The rainfall field obtained from RM captures the extremes of the true field. The maximum values in the true and RM-estimated rainfall fields are 46.24 mm and 47.57 mm, respectively. The pattern of the true field is captured with limited accuracy. The results from RM are further analyzed, which is summarized as follows:

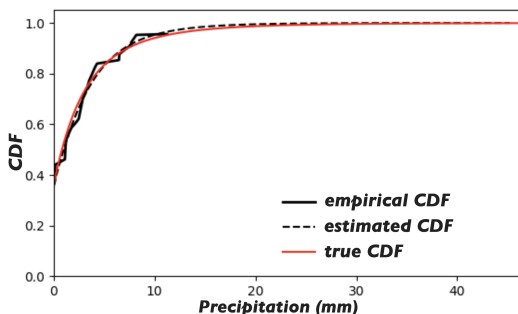

**Figure 6.** The broken and the dashed lines in black: the empirical and the estimated rainfall CDF(s) for the specific case as shown in Fig. 5 (b). The red line: the true rainfall CDF used in the generation of the true fields.

(3-1) The proposed approach is capable of capturing the extremes of the true field under the condition that the estimated rainfall CDF is relatively accurate, see the estimated and true CDF(s) for the specific case in Fig. 6.

(3-2) Unlike the estimates from ordinary Kriging, KED, or conditional merging where one obtains a Kriged mean field, from the proposed approach an infinite number of realizations for the same true rainfall field can be obtained due to the Monte Carlo framework. In Fig. 7, the mean of 100 realizations is displayed in Panel (b), compared with the true rainfall field in Panel (a). From the figure it can be seen that the mean realization is smooth and captures the pattern of the true field. By saying 'captures the pattern', it means the rain cells are more accurately located in the mean realization compared to the case in the individual realization. Yet apparently, the statistics of the mean realization (variance and covariance) are different from those of the true field. In summary, the individual realization gives relatively accurate statistics; an ensemble of such realizations gives a tendency towards the accurate locations of

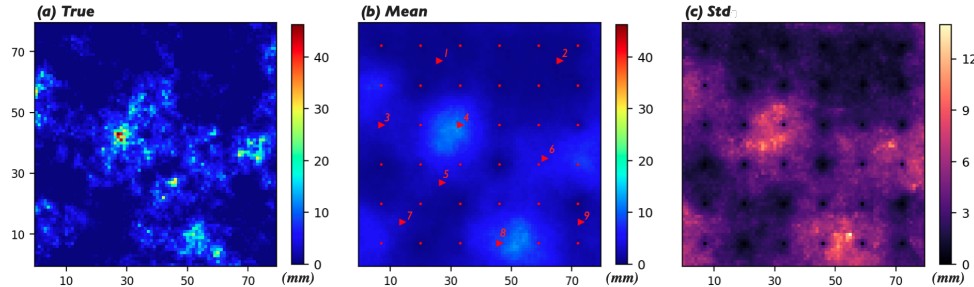

**Figure 7.** (a) The true rainfall field; (b) the mean of 100 realizations obtained from RM, where the small red dots denote the locations of the rain gauges and the red triangles together with the IDs denote the 9 pixels selected to develop box and whisker plots for in Fig. 8; (c) the standard deviation (std) of the 100 realizations.

the rainfall peaks. When feeding such an ensemble to applications such as a hydrological model, for example, one also obtains an ensemble of estimates, such that the estimation uncertainty in terms of the rainfall field propagates.

(3-3) The various estimates from RM provide a reasonable representation of the estimation uncertainty. Fig. 7 (c) displays the standard deviation (std) of the 100 realizations. The 6×6 black/zero-valued pixels which are uniformly distributed in the domain reveal the locations of the rain gauges, as all the realizations present the same values at these locations. See the exact locations of the rain gauges marked by the small red dots in Fig. 7 (b). Compared to the Kriging variance which reflects the relative position between the unknowns and the data points only, the

std map from RM is more physically meaningful. The estimation uncertainty of a pixel is affected by two factors: (a) the distance of the pixel from the data points, i.e., the uncertainty from the gauge side, and (b) the expected estimate at the pixel, i.e., the uncertainty from the radar side. The tendency is clear that the closer the pixel to the neighboring rain gauge and the smaller the expected estimate at the pixel, the lower the estimation uncertainty is at the pixel.

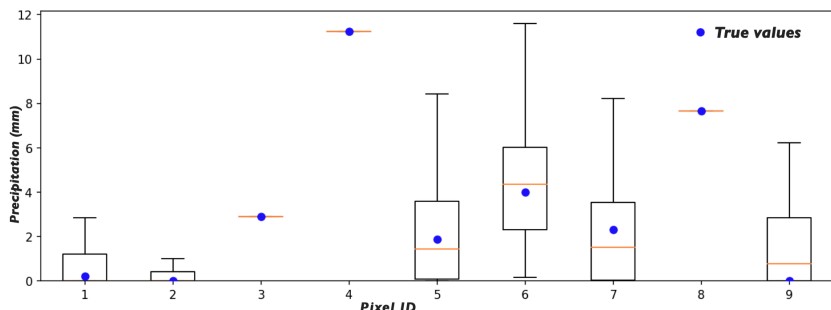

**Figure 8.** The box and whisker plots for the estimates at 9 selected pixels, see the locations and IDs of the 9 pixels in Fig 7 (b). The blue dots denote the true values, i.e., the values at the corresponding pixels in the true field.

To show the estimation uncertainty at different pixels, the box and whisker plots for 9 selected pixels are displayed in Fig. 8. The locations and IDs of the 9 pixels are given in Fig. 7 (b). It can be seen from the figure that the true values of the 9 pixels all fall in the central boxes, i.e., within the interquartile range (IQR). Among the 9 pixels, Pixels 3, 4 and 8 are collocated with the rain gauges, and all the estimates at the 3 pixels equal the true values, which demonstrates the fulfillment of the equality constraints at the rain gauge locations directly. Pixels 1 and 2 are distant

from the neighboring rain gauges, yet the two pixels are located in regions where less rainfall is expected. Thus the estimates at the 2 pixels present relatively small variance. Pixels 5, 7 and 9 are also distant from the neighboring rain gauges, yet the 3 pixels are located in regions where more rainfall is expected. Thus the corresponding estimates present higher variance. Though Pixel 6 is close to the neighboring rain gauge, it is located in a region where much rainfall is expected. Thus the corresponding estimates show relatively high variance.

## 4.2   Sensitivity Analysis

The accuracy of the estimates is mainly affected by two factors: the number of rain gauge observations and the random error in the radar estimates. To analyze the influence of the two factors, the 1000 true rainfall fields were estimated under different scenarios: radar estimates with different levels of random error (SNR = 3, 5 and 10), and rain gauge networks of different layouts (5×5, 6×6, and 7×7 rain gauges that are uniformly distributed in the domain, abbr. G25, G36, and G49, respectively).

Specifically, we focus on two statistics when evaluating the estimation accuracy: the mean and maximum of the estimated rainfall field. The former is noteworthy when the estimated rainfall field is used in studies where the water balance of a region is important; the latter is of great concern when the extreme of the region is of interest, for example, for stormwater management, flood risk assessment, etc.

### 4.2.1   Field maximum

The maximum of the rainfall field obtained from RM or KED is compared with the true maximum. As literally by RM one could obtain an infinite number of realizations for each of the 1000 true rainfall fields, we simulate an ensemble of 20 realizations for each true field and use the median of the 20 errors - the difference between the simulated and the true maxima for each realization - as the representative. Fig. 9 shows the histograms of the errors w.r.t. the estimated field maxima for the 1000 fields by RM and KED. Specifically, the upper panel shows the influence of the layout of rain gauges, and the lower panel shows

the influence of the random error in the radar estimates. For the sake of clarity, we only display the scenarios with the most and least rain gauges (i.e. G25 and G49) in the upper panel, while the random error in the radar estimates is fixed at SNR = 5. Similarly, in the lower panel, only the results of the scenarios with the largest and smallest random error in the radar estimates (i.e. SNR = 3 and 10) are displayed, while the layout of rain gauges is fixed at G36. It can be seen from the figure that negative biases exist in all the results from KED, while the biases of the results from RM are not obvious. Further, it can be seen that

increasing the number of rain gauges and reducing the random error in the radar estimates are both beneficial to improve the quality of the estimates, i.e., to reduce the size of the error and to decrease the estimation variance (to shrink the scattering of the histogram).

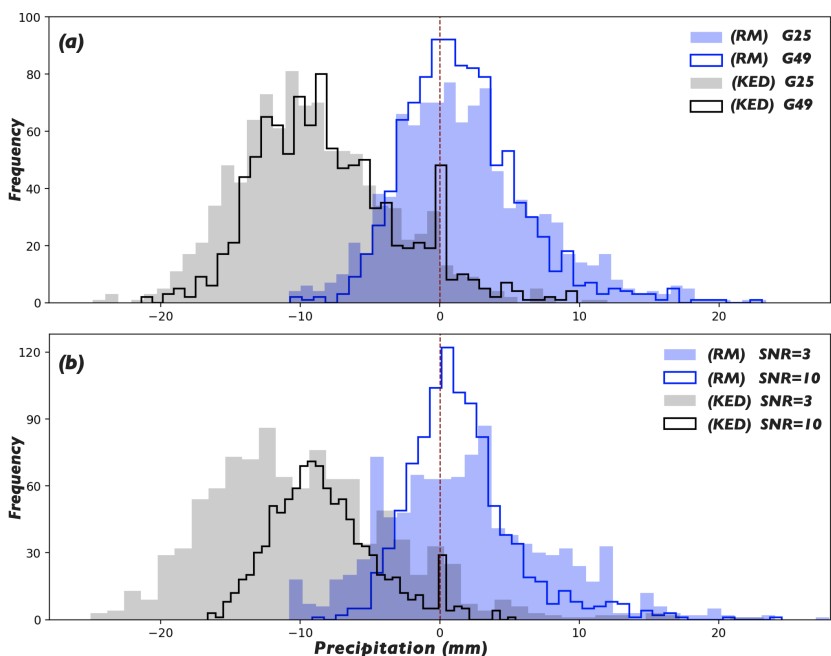

**Figure 9.** The histograms of the errors w.r.t. the estimated field maxima for the 1000 true rainfall fields by RM and KED. (a) The results of the scenarios with different layouts of rain gauges, while the random error in the radar estimates is fixed at SNR = 5. (b) The results of the scenarios with different random error in the radar estimates, while the layout of rain gauges is fixed at G36.

**Table 1.** The mean error (ME) and the interquartile range (IQR) of the errors w.r.t. the estimated field maxima for the 1000 true rainfall fields by RM (the three columns on the left), and by KED (the three columns on the right). The column name denotes the SNR of the radar estimates and the row name denotes the layout of rain gauges.

| ME | 3 | 5 | 10 | 3 | 5 | 10 |
|-----|-------|-------|-----------|---------|--------|-------------|
| G25 | 2.182 | 1.929 | 1.611 | -10.114 | -9.104 | -8.741 |
| G36 | 1.819 | 1.698 | ***1.451*** | -9.760 | -8.700 | -8.250 |
| G49 | 1.740 | 1.611 | 1.538 | -8.997 | -7.960 | ***-7.620*** |

| IQR | | RM | | | KED | |
|-----|-------|-------|-----------|-------|-------|-----------|
| G25 | 9.551 | 6.497 | 4.798 | 9.650 | 7.166 | 5.312 |
| G36 | 7.742 | 5.574 | ***4.218*** | 8.703 | 6.539 | ***4.624*** |
| G49 | 6.665 | 5.057 | 4.242 | 8.652 | 6.684 | 5.101 |

The best performances in terms of the ME and IQR for both methods are printed in bold and italic.

The histogram shown in Fig. 9 can be summarized by two statistics: mean error (ME) and interquartile range of the errors (IQR, the range between the quantiles 0.75 and 0.25). Table 1 shows the two statistics evaluated from the results of all the

scenarios. It can be seen from the upper part of the table that slight overestimation exists in the estimates from RM, and relatively significant underestimation exists in the estimates from KED. For both methods, increasing the number of rain gauges and reducing the random error in the radar estimates both help in reducing the ME and shrinking the IQR. Yet for RM, it is not necessarily the 'more' the better. For example, the best performances in terms of the ME and IQR both present in the scenario (SNR=10, G36). The scenario (SNR=10, G49), which is assumed to perform the best, is only ranked the second, which might be triggered by the over-fitting problem in the estimation of the rainfall CDF. When the radar estimates are relatively accurate, a certain number of rain gauges is enough to sample sufficient information. Increasing the number of rain gauges further can lead to the over-fitting problem due to the surplus of information.

### 4.2.2  Field mean

The estimated mean of the rainfall field by RM or KED is compared with the true mean to evaluate the accuracy in terms of the mean statistic. For each of the 1000 true fields, an ensemble of 20 realizations is produced by RM and the mean of the 20 errors - the difference between the simulated and the true means for each realization - is used as the representative. Fig. 10 shows the histograms of the errors w.r.t. the estimated field means for the 1000 fields obtained from RM in Panels (a, b) and from KED in Panels (c, d). It can be seen from the figure that compared to RM, KED seems to be more sensitive towards the two factors. Further, slight positive biases can be observed in the estimates from KED for all the displayed scenarios. Among the four sub-figures, (a) and (c) show the influence of the layouts of rain gauges for RM and KED, respectively; (b) and (d) show the influence of the random error in the radar estimates. For KED the influence of the two factors is ambiguous. One could say increasing the number of rain gauges or reducing the random error in the radar estimates helps in reducing the estimation variance, yet one can hardly say it helps in reducing the size of the error. For RM the influence of the two factors is not obvious, and the biases in the estimates are not obvious either.

We still use the two statistics - ME and IQR - to generalize each histogram, and the results of all the scenarios are shown in Table 2. It can be seen from the upper part of the table that RM outperforms KED in terms of the ME, as the ME(s) of all the scenarios from RM are smaller than the best performance of KED. It is worth considering that from the previous analysis, KED is expected to underestimate the maximum of the rainfall field, yet the results shown here indicate that KED overestimates the mean of the field. Thus KED is likely to overestimate the lower quantiles of the rainfall field. Further, it is demonstrated again by the table that RM is not very sensitive towards the two factors as one can barely see any tendency.

## 5  Discussion

### 5.1  The two cores of the approach

In this paper, an approach to obtain estimates of spatial rainfall fields together with the associated uncertainty is proposed. The approach has two cores: the method to compute the rainfall CDF (referred to as CDF method hereafter) and the method random mixing (RM) to simulate spatial random fields under constraints.

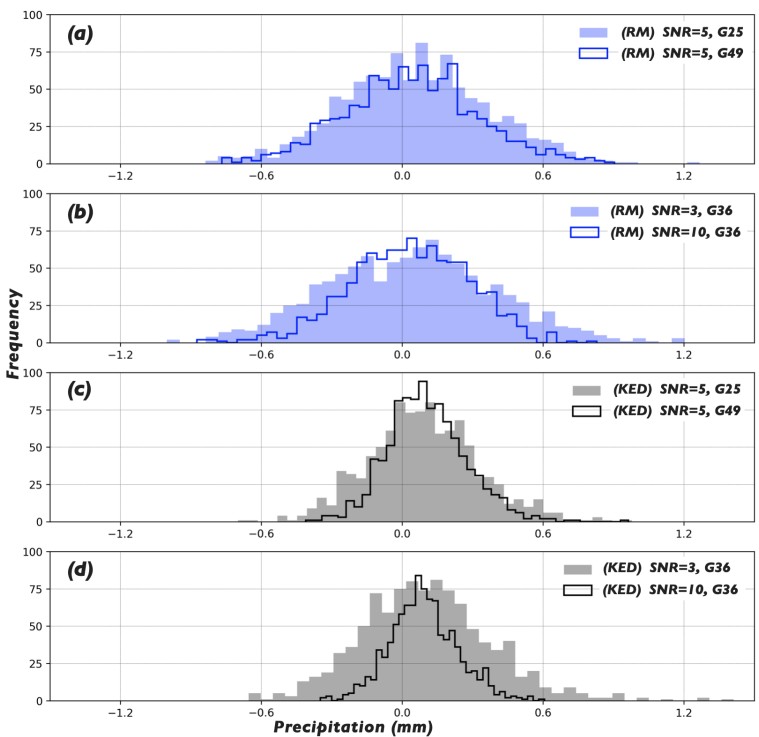

**Figure 10.** The histograms of the errors w.r.t. the estimated field means for the 1000 true rainfall fields by RM (a, b) and KED (c, d). The estimates are obtained under different scenarios, see the texts in the sub-figures for the specific scenarios.

**Table 2.** The mean error (ME) and the interquartile range (IQR) of the errors w.r.t. the estimated field means for the 1000 true rainfall fields by RM (the three columns on the left), and by KED (the three columns on the right). The column name denotes the SNR of the radar estimates and the row name denotes the layout of rain gauges.

| ME | 3 | 5 | 10 | 3 | 5 | 10 |
|---|---|---|---|---|---|---|
| G25 | 0.049 | 0.055 | *0.023* | 0.103 | 0.090 | *0.082* |
| G36 | 0.038 | 0.033 | 0.029 | 0.105 | 0.096 | 0.092 |
| G49 | 0.038 | 0.039 | 0.030 | 0.116 | 0.106 | 0.099 |

| IQR | | RM | | | KED | |
|---|---|---|---|---|---|---|
| G25 | 0.545 | 0.412 | *0.340* | 0.434 | 0.300 | 0.212 |
| G36 | 0.490 | 0.372 | 0.352 | 0.355 | 0.251 | 0.182 |
| G49 | 0.438 | 0.362 | 0.353 | 0.294 | 0.214 | *0.154* |

The best performances in terms of the ME and IQR for both methods are printed in bold and italic.

The CDF method is the foundation of the approach. A resultant rainfall CDF with sufficient accuracy is necessary for a successful application of the approach. The statistics of intermittent precipitation are non-Gaussian, and such properties restrict the usage of well-established stochastic models that assume Gaussianity (Pulkkinen et al., 2019). Specifically in this study, the rainfall CDF is important in two respects: (a) it is used in the data transformation whereby the simulated Gaussian fields are transformed to rainfall fields of interest; (b) it is used in the definition of the constraints in terms of the normalized Gaussian marginal.

RM, on the other hand, is an excellent tool that performs conditional simulation in Gaussian space, yet it is not irreplaceable. Another conditional simulation method could have been used, for example, phase annealing (Yan et al., 2020). RM is employed in this study due to (a) the relatively high efficiency which makes mass production of realizations possible, and (b) code availability - a Python package for conditional simulation of spatial random fields using RM is available where the authors give practical demonstrations on the application of the method (Hörning and Haese, 2021).

## 5.2   The scope of the approach

The proposed approach is aimed at estimating spatial rainfall fields of short accumulation time: 15 min, 10 min, or even 5 min. The temporal aspects of QPE (Quantitative Precipitation Estimates) are outside the scope of this study. Unlike the acquisition of QPF (Quantitative Precipitation Forecasts) by a radar-based nowcast model, for example, where modeling of the temporal evolution of the precipitation field is of interest, in this study the spatial rainfall fields are obtained in a hindcast mode. Yet the approach has great potential to improve the quality of QPF. As has been proposed by Shehu and Haberlandt (2020) that the predictability loss of a nowcast model is mainly caused by the inability of radar to capture the true rainfall field and because the Lagrangian Persistence is unable to model the temporal evolution of the rainfall field. The approach could therefore be used to improve the rainfall field fed into the model.

Furthermore, the skill of radar-based nowcasting has been experiencing an evolution from the deterministic to the probabilistic framework to estimate the predictive uncertainty (e.g., Pierce et al., 2012; Shehu and Haberlandt, 2020). A common approach is based on stochastic simulation in which correlated noise fields are used to perturb a deterministic nowcast (e.g., Liguori and Rico-Ramirez, 2014; Foresti et al., 2016). The output of the proposed approach - an ensemble of estimates that considers both radar and rain gauge data - could be used to perturb a deterministic model, and this ensemble of estimates is more hydrologically meaningful than a random perturbation, as used in many probabilistic nowcast models (e.g., Bowler et al., 2006; Berenguer et al., 2011; Pulkkinen et al., 2019). Besides, precipitation exhibits variability in space; hence it is challenging to estimate spatial rainfall fields in a deterministic manner, even in a hindcast mode. If the nowcasting community can embrace a change from the deterministic to probabilistic, a similar change could also happen in the hindcasting community.

## 5.3 Concerning the CDF method

### 5.3.1 Basic assumption

The rainfall CDF is computed from a set of rain gauge observations and the radar estimates. The basic assumption of the method is that the radar estimates can represent the pattern of the rainfall field to a reasonable extent. Random error which can reduce the representativeness of the radar estimates is accounted for, and the quality control steps in the algorithm are specifically designed to get rid of the negative effects from random error. While the degradation of the representativeness due to the systematic effects in the radar estimates, e.g., range-dependent errors associated with the increasing height of the radar beam or increasing sampling volume as one goes away from the radar antenna, is not accounted for. Thus quality control to get rid of the systematic effects in the radar estimates is necessary.

### 5.3.2 Impact from distribution and size of rain gauge observations

The capability of the CDF method is affected by the distribution and size of rain gauge observations. A uniform distribution of the rain gauge observations is not required, yet the gauge observations should not be too clustered, as it is normally considered that one has a better chance to obtain spatially representative samples from a relatively evenly distributed rain gauge network.

For mesoscale hydrological studies often only small sample sizes of irregularly distributed rain gauge stations are available. To obtain a rainfall CDF with adequate accuracy, enough gauge-radar pairs (samples) should be available. There are two possibilities to increase the sample size:

1) Increase the sample size in space. For small domains with only a few rainfall stations (say 10), it can be assumed that rain parcels move uniformly. Under this assumption, one can displace the radar quantile map using a vector that decreases the radar-gauge mismatch (the Spearman's rank correlation as the measure, for example), then refind the gauge-respective quantiles in the displaced quantile map. The result is 10 new pairs $(r_k, u'_k)$ for $k = 1, \ldots, 10$, where $u'_k$ is the refound radar quantile. Normally a stack of such vectors (N) can be found, which results in $10 \cdot N$ new samples. Surely one should limit the domain size to apply the above practice, and the practice has been applied in the work by Yan and Bárdossy (2019) to find the empirical rainfall CDF for a domain with the size $19 \times 19$ km$^2$.

2) Increase the sample size in time. A fixed time window can be set by assuming a relatively stable distribution during the relevant time interval, and the gauge-radar pairs in the time window can be used in the calculation of the rainfall CDF.

A combination of both practices can enrich the sample size to a remarkable extent.

### 5.3.3 Impact from spatial intermittency

In this paper, the performance of the CDF method has been tested on a hydrologically interesting spatial intermittency $u_0 = 0.36$. Practically, the choice of $u_0$ has a minor influence on the performance. When $u_0 > 0.36$ (i.e. a larger dry-area-ratio), the point where the rainfall CDF intersects the y-axis moves up, and also there are more zero-samples in both the radar and rain

gauge data. When $0 < u_0 < 0.36$ (i.e. a larger wet-area-ratio), the point moves down and there are fewer zero-samples. The method is problematic if $u_0 = 0$, i.e., the entire domain is wet. In that case, there is no $(0, u_0)$ and the enforcement in Step 4 (Sect. 2.1) that the CDF passes through the point $(0, u_0)$ should not be applied then.

### 5.3.4    Applicability in terms of the spatial scale

The CDF method is valid at a limited spatial scale. At a point, the domain must be too large for a single CDF to represent all
processes. This limit should be related to factors such as rainfall regime, topography, geography, etc. Yet in the experimental context of this study, practically significant information on the limit of the spatial scale cannot be provided. A further study based on realistic datasets is therefore required.

## 6    Conclusions and Outlook

In this paper, an approach to simulated spatial rainfall fields conditioned on radar and rain gauge data is proposed. The ap-
proach has two cores: the method to compute the marginal distribution function of the rainfall field and random mixing which conducts conditional simulation in Gaussian space. An artificial experiment has been made to test the efficiency of the proposed approach, and the results are compared with those from 3 commonly seen Kriging methods: ordinary Kriging, Kriging with external drift (KED), and conditional merging. The proposed approach outperforms KED and conditional merging especially in the estimation of the extremes. The estimates by the proposed approach differ from those by the others in two respects.
First, unlike the Kriging methods where a Kriged mean field is given by minimizing the estimation variance, the output of the approach is an ensemble of estimates (realizations) with identical statistics (mean, variance, correlation function, etc.) due to the Monte Carlo framework. The individual realization strictly fulfills the equality constraints at rain gauge locations and presents a field pattern that is similar to the radar estimates, and an ensemble of such realizations provides a tendency towards the accurate locations of the rainfall peaks. Second, the various estimates from the proposed approach provide a reasonable
representation of the estimation uncertainty. In addition to representing the relative positions between the unknowns and the data points - the information and the associated uncertainty (due to the sparsity) from rain gauge data - as what the Kriging variance could also provide, the estimation uncertainly is more physically meaningful by considering the information and the associated uncertainty from radar data as well.

     Further, the sensitivity of the proposed approach towards the two factors - the number of rain gauges and the magnitude of
random error in the radar estimates - has been analyzed. Specifically, we focus on the accuracy of the approach in estimating the maximum and the mean of the rainfall field, and the results are compared with those from KED. Concerning the estimation of the maximum, increasing the number of rain gauges and reducing the random error in the radar estimates are both helpful to improve the estimation quality. Yet when the radar estimates are relatively accurate, a certain number of rain gauges is already enough to sample adequate information of the rainfall field. Increasing the number of rain gauges further can lead to
the over-fitting problem in the estimation of the rainfall CDF. Comparing the two methods, the proposed approach outperforms KED in terms of the mean error (ME) and the interquartile range (IQR) of the errors. Concerning the estimation of the mean,

the proposed approach is not as sensitive as KED towards the two factors. The proposed approach outperforms KED in terms of the ME, yet the estimation variance is generally larger than that from KED.

In this paper, we present a simulation study where synthetic rainfall fields were used as the true fields, and radar and rain
gauge data used for the application of the proposed approach were derived therefrom. Due to the full control over the stochastic process, the accuracy of the estimates could be examined comprehensively on the overall statistics and the sensitivity of the approach could be analyzed as well. However, there are several practical questions that cannot be answered without the relevant investigation based on realistic datasets. For example, explicit information on the limit of the spatial scale to apply the CDF method is required and the effects of small size and/or irregular distribution of rain gauges on the performance of the approach
remain interesting topics. Thus further studies based on realistic datasets are necessary.

*Author contributions.* The first author, JY, did the programming work and part of the manuscript writing. The first author, FL, did lots of computational work and part of the manuscript writing. They contributed equally to this work. The second author, AB, contributed to the research idea and supervised the research. The third author, TT, provided valuable suggestions for the revision of this article.

*Competing interests.* The authors declare that they have no conflict of interest.

*Acknowledgements.* This research has been supported by China Postdoctoral Science Foundation (Grant No. 0400229136) and the National Natural Science Foundation of China (Grant Nos. 04002340416, 51778452, and 51978493). The authors are grateful to the editor and the reviewers of this paper (Nadav Peleg, Remko Uijlenhoet, Scott Sinclair, and one anonymous reviewer) for their excellent critical reviews and valuable suggestions which improved the manuscript substantially.

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
