# Peer review of "Conditional simulation of spatial rainfall fields using random mixing: A study that implements full control over the stochastic process"

_Hydrology and Earth System Sciences, 2021_

## Referee Comment (RC2)

**Review of manuscript hess-2021-56, "Simulation of rainfall fields conditioned on rain gauge observations and radar estimates using random mixing" by Jieru Yan et al.**

Remko Uijlenhoet

*General remarks*

Quantitative Precipitation Estimation (QPE), i.e. our ability to estimate space-time rainfall fields using data collected by weather radars and rain gauges, is important for obtaining hydrological process understanding as well as for water management applications. As the space-time sampling properties and other instrumental characteristics of radar and gauges are complementary to some extent, several methods have been proposed in the scientific literature to optimally combine ("merge") radar and gauge data to obtain spatial precipitation estimates over areas such as river catchments.

Conditional simulation has great potential to be used in spatial precipitation estimation as it is able to account for uncertainties associated with QPE through a Monte Carlo framework. The authors claim that one obstacle hampering the application of conditional simulation in QPE is obtaining the marginal distribution function of the rainfall field with sufficient accuracy. They propose a method to obtain this marginal distribution function based on rain gauge observations and radar estimates, namely Random Mixing (RM). Spatial rainfall fields estimated using RM are compared with those from well-known radar-gauge merging methods: Ordinary Kriging, Kriging with External Drift, and Conditional Merging.

The subject of radar QPE through radar-gauge merging is definitely suitable for publication in HESS. However, apart from the specific remarks and editorial remarks provided below, this study has two important limitations, which should be stressed more clearly in the paper: 1) it is a pure simulation study – in spite of what is suggested by the title of this paper, there are no radar or rain gauge data employed in this study; 2) the study only considers the estimation of *spatial* rainfall fields, completely neglecting the *temporal* aspect of QPE, which is so important for hydrological applications. These two limitations should be reflected more clearly in the title, abstract and introduction of the paper, as far as I am concerned. Moreover, a proper Discussion section appears to be missing from the paper.

Overall, this paper proposes an interesting and relevant stochastic radar-gauge merging method, which is quantitatively tested in a controlled environment through stochastic simulations. Although I found the presentation of the proposed algorithm sometimes challenging to follow (see remarks below), I do find it an interesting and relevant addition to the hydrological engineering literature. Thanks very much in advance for taking my suggestions as provided above and below into account.

*Specific remarks*

- L.45: Hitschfeld and Bordan (1954) is a classical reference for (physical) radar attenuation correction. I would not cite this reference as a typical example of (statistical) mean field bias correction. Perhaps you could refer to some of the papers by D.-J. Seo (formerly at NOAA-NWS) on this issue.
- L.91–97: If I understand correctly, you are forcing the intermittency in the (transformed) radar data to be the same as that of the rain gauge data. However, since both sensors (radar and gauges) have very different space-time sampling properties, the probability of finding zeroes in the radar rainfall field may be quite different from the probability of finding zeroes in the rain gauge field. In particular, the spatial aggregation associated with radar rainfall observations and the temporal aggregation associated with rain gauge observations make that their respective capacities to detect intermittency in space and time are quite different. To what extent does this affect your approach?
- L.118–124: Here, "correlation function" and "variogram" are used interchangeably. Please choose one of the two terms as a measure of spatial dependence. Moreover, it may be true that "it can be seen that the empirical and the true variograms have very similar patterns", but that does not mean that the truncated Gaussian fields necessarily have the appropriate (i.e. a realistic) spatial intermittency structure. Is that a problem for your approach or not? If not, why not?

- L.192: "George Dantzig's simplex algorithm, the BFGS method, etc." – please add literature references to these methods.
- L.135–204: I am able to follow the general reasoning of Section 2.3 ("Random Mixing"), however, I am not able to grasp all the intricacies of the proposed method. It would be good if one of the referees would be familiar with the RM method, or at least if he/she would have a solid background in stochastic processes.
- L.212–213: The authors use again the notion of the correlation function (also see Fig. 3 and caption), whereas previously they employed the variogram to express the degree of spatial dependence. For the sake of consistency, it would be good to stick with the notion of variogram and use that throughout the paper. Alternatively, the correlation function could be employed instead of the variogram. However, please do not use these notions interchangeably.
- L.217–218: "rainfall rates over an area, or the objective of this paper: short time rainfall over an area" – what do you mean exactly with "short time rainfall"? To what extent is it different from "rainfall rates"?
- L.232–233: "A random error is therefore introduced to mimic this kind of error", namely "factors […] that can reduce the representativeness of the radar estimates for the rainfall pattern on the ground, such as evaporation, complex terrain effects and anthropogenic influences" (L.231–232). However, many of the representativeness errors in radar rainfall estimation are not random, but systematic, e.g. range-dependent errors associated with the increasing height of the radar beam above the ground the further one goes away from the radar antenna, or the associated increase of the radar sampling volume. How would one account for such systematic effects in the proposed simulation? How important would it be to incorporate such systematic effects?
- L.238 – 240: "It should be notified that the introduction of random error in Step 1 differs the radar estimates from the true rainfall field in terms of the field pattern, yet the changes in the statistical properties are tiny" – see previous remark: would these "changes" still be "tiny" if more realistic *systematic* errors would be accounted for in the simulation framework, rather than *random* errors alone?
- L.252–253: "The choice of the two parameters - factor 0.87 and exponent 0.83 - is quite arbitrary. We have modeled a case when radar underestimates the precipitation. Surely one could model other cases." – this approach does not appear to be based on sound assumptions ("quite arbitrary"). How did you determine these parameter values? Have you considered other values? What process are you trying to mimic here, the Z–R relation? With the typical Z–R exponent of 1.6, one would perhaps have expected an exponent of 1 / 1.6 = 0.625, rather than 0.83. In any case, a clear motivation for using a value of 0.83 seems to be lacking. The same holds for the employed value for the factor (0.87). In addition, how representative are your results, if these parameter values were chosen in such an arbitrary manner?
- L.265: "This gives an approximate coverage of one rain gauge for every 256, 178, and 131 km$^2$, respectively." – why these rain gauge densities? How do they compare to rain gauge densities encountered in practice? Please motivate these choices, which (again) seem to be quite arbitrary.
- Fig. 5: These are certainly interesting simulation results, however, the generated random fields do not really resemble actual radar-estimated rainfall fields, which often display a strong sense of directionality (anisotropy), associated with the prevailing movement direction of the rainfall field. So, how representative do you think your simulation results are for practical purposes, e.g. when applied to radar and rain gauge data from the German national weather service (DWD)? This also refers to the lacking temporal aspect of radar QPE, which was mentioned already under "general remarks".
- L.295–296: "to a certain degree" – to what degree? Please try to be as concrete as possible.
- L.306–307: "Instead, one can obtain an infinite number of realizations for the same true rainfall field by RM" – but Kriging and other geostatistical methods can also be employed for (conditional) simulations, can't they? Or would these methods lead to the issues associated with the inaccurate representation of the marginal distribution of the generated rainfall fields, which the authors referred to in the abstract and introduction of their paper? If yes, please mention this explicitly.
- L.314: "the mean realization is more helpful in identifying the locations of the rain cells" – OK, but this apparently goes at the expense of a realistic spatial variability. Can you get the locations *and* the variability right simultaneously with the proposed method? That would be relevant for practical applications, it seems to me.
- Fig. 9, Table 1: Have "G25" (presumably 5x5 rain gauges), "G36" (6x6) and "G49" (7x7) been defined before? If not, please do so.

- L.405: After presenting the Sensitivity Analysis (Section 4.2), which is part of the Results (Section 4), you immediately jump to the Conclusions. However, a true Discussion section, where one puts one's own results into perspective, by critically discussing assumptions and the associated limitations and by comparing obtained results with results reported elsewhere in the scientific literature, appears to be lacking. However, this is an important element of any scientific study. Therefore, I urge the authors to include such a Discussion in a revised version of this paper (unless I missed it).

*Editorial remarks*

- L.1, 19: "temporospatial" → "spatiotemporal".
- L.5, 38: "lack" → "lacks".
- L.10, 62, 81: "accuracy" → "sufficient accuracy".
- L.43, 48: "generic" → "class".
- L.43: "have" → "has".
- L.60: "hamper" → "hampers".
- L.78: "cdf" → "CDF" (here and elsewhere in the paper).
- L.80: "cdf-rainfall" → "rainfall CDF" (here and elsewhere in the paper).
- L.84: replace full stop (".") after "as follows" with colon (":").
- L.85–86: "the pixels" → "pixels".
- L.87: "Uniform" → "Transform".
- L.88, 90, 254: "uniformed" → "transformed".
- L.99: "Due to" → "For".
- L.102: "distribute unbiasedly" → "are distributed without bias".
- L.107: "in the marginal" → "in terms of the marginal" (if this is what you mean).
- L.109: "In specific" → "In particular" / "Specifically".
- L.109: replace full stop (".") after "constraints" with colon (":").
- L.110, 116: "in the standard normal marginal" → "in terms of the standard normal marginal" (if this is what you mean).
- L.118: "A problem" → "The problem".
- L.119: "namely" → "where".
- L.119: "And due" → "Due".
- L.125: "to a certain degree" → "as close as possible".
- L.129–140: Remove "the formula of the".
- L.137: "an estimate for" → "an estimate of".
- L.141–142: "distribution" → "distributions".
- L.142, 180: "L-2" → "L2".
- L.145: "(Bárdossy and Hörning, 2016)" → "Bárdossy and Hörning (2016)".
- L.146: "function, and we …" → "function. We …".
- L.157: "subject" → "subject to".
- L.159: "equation" → "equations".
- L.159: "it" → "this".
- L.161: Insert "L2" before "norm".
- L.162, 169, 237: "subject" → "imposed" (if that is what you mean).
- L.191: Remove "the" at the end of this line.
- L.195–196: "as the following" → "as follows".
- L.209: "is representative" → "is assumed to be representative".
- L.224: "Namely, …" → "Hence, …".

- L.228: replace full stop (".") after "applied" with colon (":").
- L.239: "differs" → "changes" / "affects".
- Figs. 4, 9, 10: The y-axis labels read "number of frequencies". This should simply be "frequency" (which means "number of occasions").
- L.260: If you use "gauge" rather than "gage", you should also employ "ungauged" instead of "ungaged".
- L.260: "from the rain gauge" → "to the rain gauge".
- L.271: "in details" → "in detail".
- L.272, 273, 308, 385: "on Panel(s)" → "in Panel(s)".
- L.278, 280, 283, 294, 297: "extreme" → "extremes".
- L.282: "scenario is radar" → "radar scenario" (if this is what you mean).
- L.296, 411: replace full stop (".") after "as follows" with colon (":").
- L.310: "capture" → "captures".
- L.317: "uniformly distribute" → "are uniformly distributed".
- L.322: "from" → "to".
- L.330, 332, 333: "sited" → "located".
- L.336, 341, 342, 343: do you mean "extreme" or "maximum"?
- L.338: Insert "for" before "stormwater management".
- L.349, 352, 422–423: "field extreme" – do you mean "field maximum"?
- L.391–392: "And for …" → "For …".
- L.437 – 532: Please carefully check your reference list. Some articles titles are capitalized, others not; book titles need to be capitalized, whereas some are not; the price of Battan (1973) does not have to be included in the reference (L.444).

---

## Author Comment (AC5)

We thank Scott Sinclair for the comments for improvement. In the following, we present our replies on the general comments and the detailed and editorial comments.

**Concerning the general comments**

Surely there are many interesting questions to be discussed concerning the applicability and capability of the proposed method, for example, on what conditions does the proposed method allow a better estimation of the rainfall field and the uncertainty of the estimation than the established Kriging methods. The other referee - Prof. Uijlenhoet - has indicated that a proper discussion section is missing in the manuscript, where one could discuss the assumptions and the associated limitations of the proposed method. We think here the referee implicitly raises the same problem.

It would indeed be of value to understand whether this technique has better performance over a range of time/space scales than other methods. And it is correct that the proposed method fixes both the marginal distribution function and the spatial correlation for each field, while for Kriging only the spatial correlation is fixed. However, this is a pure simulation study, where one has full control over the stochastic process (i.e., knowing the true rainfall fields). Under this context, even if we present better performance of the proposed technique over a range of time/space scales than other methods, the persuasiveness is limited, as synthetic data can only partially represent the reality. Yet as described in AC2, the goodness of a pure simulation study is that one has full control, thus can verify the accuracy of the estimates more comprehensively, compared to the case when the true datasets (radar and gauges) are used, where one only verifies the accuracy of the estimates at limited locations using, e.g., leave-n-out cross-validation. The capability of the proposed method is demonstrated in a synthetic experiment, and many realistic questions are to be answered; hence a further study based on realistic datasets is required.

**Concerning the detailed and editorial comments**

The referee has listed 11 such comments, among which, the 1st, 4th, 7th are editorial comments which require no replies, yet the relevant improvements should be made. In the following we present our replies to the other comments:

**The 2nd comment**: pg 4, line 95 - is this quality control step justified by any reason other than practical considerations of the method?

The quality control steps are applied to rule out the negative effects of the random error introduced in the radar estimates, which can lead to inconsistency at zeros, as well as the Spearmans rank correlation of the two datasets (gauge observations and the collocated radar quantiles) not being exactly 1. The quality control steps are effective in eliminating the interference from the random error, while for systematic errors, e.g., the range-dependent errors, the effectiveness of the quality control steps is limited.

**The 3rd , 6th and 8th comments** – concerning the algorithm to compute the marginal distribution of the rainfall field

The 3rd comment: We fully agree that at some point the domain must be too large for a single CDF to represent all processes, namely, the spatial CDF can be considered valid at a limited spatial scale, and this scale should be related to factors such as the rainfall regime, local climate, topography, etc. The referee has raised a very interesting question that we have not considered previously. In the experimental context of this study, we could not answer this question properly, and a further study based on realistic datasets is therefore required.

The 6th comment: the choice of the intermittency $u_0$. Practically, the choice of $u_0$ should not change the results significantly in the experimental context of this study. We have tested a hydrologically interesting case, $u_0 = 0.36$. When $u_0 > 0.36$ (larger dry-area-ratio), it only means the starting point of the CDF (the intersecting point on y-axis in Fig. 3b) moves up, and there are more zero-samples in both datasets (gauge and radar). When $0 < u_0 < 0.36$ (larger wet-area-ratio), the starting point of the CDF moves down, and there are fewer zero-samples in both datasets. The algorithm will be problematic if $u_0 = 0$, i.e., the entire domain is wet. In that case, there is no $(0, u_0)$, and thus there

should be no enforcement that the CDF intersects at the point $(0, u_0)$ when fitting a theoretical CDF (Line 104).

The 8th comment: the referee has raised a similar question here as the other referee, see RC1 the 3rd comment, "In practice, for mesoscale hydrological studies only small sample sizes of irregular distributed recording rainfall stations are available...". To obtain an accurate CDF using the proposed method, a certain number of rain gauge observations should be given, which are not necessarily uniformly distributed, yet the rain gauge observations should not be too clustered. And even with a small sample size, there are possibilities to improve the applicability of the method, see the reply to the 3rd comment in AC1.

**The 5th comment**: Lines 163, 164 - how to decide to increase N?

Take the case when the number of rain gauge observations is 25 as an example. We have 25 linear constraints, $K = 25$ in Eqn. (7). One should choose an initial value of $N (N > 25)$ - the number of unconditional random fields - say 50. Then solve the under-determined system defined in Eqn. (7), and find the set of weights with the least norm, $\sqrt{\sum_{i=1}^{N} \alpha_i^2}$, using, e.g., the singular value decomposition. If the norm is above a certain threshold (say 0.1), then increase N by a step of 10, for example, and solve Eqn. (7) again. Repeat the above procedure until the norm $\leq 0.1$.

**The last 3 comments** – concerning the error statistics in estimation of the extremes.

What is shown in Fig. 9, Line 367, and Table 1 is the error statistics in estimating the single extreme for each field. Yet it should be clarified that when converting the simulated Gaussian field $Z$ to rainfall field $R$ using the normal-quantile transformation, Eqn. (15) in the manuscript

$$R = G^{-1}(\Phi(Z))$$

all pixels satisfying $Z(x) > \Phi^{-1}(0.995)$ are converted to $G^{-1}(0.995)$ to get rid of the numerical effects, where $G(\cdot)$ and $\Phi(\cdot)$ are the CDFs of the rainfall field and the standard normal distribution, respectively. From this perspective, one could say the results shown are the extremes above a threshold quantile, though an extremely high threshold. If a lower threshold is used (say 0.95, 0.9, 0.8), the bias in the results from both RM and KED should decrease. And in that case, the contrast of the results from the two methods should not be as significant as shown in Fig. 9, Line 367, and Table 1. In a nutshell, the higher the threshold quantile, the more significant the contrast is.

The last comment: as for whether to include the "extremes of errors" in Table 1. We think the histogram of the errors provides a very intuitive expression of how the errors distribute, and we have shown the most extreme cases in Fig. 9: the scenarios that are most favorable and unfavorable for the methods. Table 1 is derived from the histograms and provides supplementary information on the means and scatterings of the histograms. As for the extreme errors (the referee meant the maximum absolute error if we understand correctly), it is our opinion that this statistic seems not so informative, and the two methods seem to have comparable performances, as observed from Fig. 9.

---

## Author Response (AR1)

**To Anonymous Referee**

- **Question**: The unusual definition of a quantile should be made clear before use or the classical definition non-exceedance probability should be used.

  **Answer**: We have used 'quantile' instead of the classical nomenclature 'non-exceedance probability' in the description of the algorithm of the CDF method. And it has been specified in L.89 in the revised manuscript.

- **Question**: The introduction suggests that the estimation of precipitation in high spatio-temporal resolution is important. The paper only deals with spatial simulation. There is no reference to the simulation of time series of precipitation. How can this be reached? This should be at least discussed briefly.

  **Answer**: The title of this paper has been modified to reveal the focus of this study - conditional simulation of spatial rainfall fields. The temporal aspects of QPE (Quantitative Precipitation Estimates) are outside the scope of this study. Unlike the acquisition of QPF (Quantitative Precipitation Forecasts) by a radar-based nowcast model, for example, where modeling of the temporal evolution of the precipitation field is of interest, in this study the spatial rainfall fields are obtained in a hindcast mode. Given the observed radar estimates and station data (some weather condition that has already existed), we try to estimate the true rainfall field. The scope of the proposed approach has been discussed in Sect. 5.2 (pg 20) in the revised manuscript where we also discuss the potential of the approach to improve the quality of QPF.

- **Question**: In practice, for mesoscale hydrological studies often only small sample sizes of irregular distributed recording rainfall stations are available (e.g. about 10 stations). How uncertain is the estimation of the CDF with only a few point pairs of data? What is then the value of radar data as additional information to build the CDF?

  **Answer**: In Sect. 5.3.2 (pg 21) in the revised manuscript, we discuss two possibilities to increase the sample size in space and time, so as to improve the applicability of the proposed CDF method. At the end of the revised manuscript (L.499-500), we also indicate that the effects of small size and/or irregular distribution of rain gauges on the performance of the proposed approach should be further investigated based on realistic datasets.

  As for the value of radar data in the algorithm to build the rainfall CDF, the radar data provides a hint on the representativeness of the rain gauge data. For example, has the extreme of the rainfall field been properly sampled by the gauges? If not, to what extent has the extreme been underestimated by the samples (rain gauge observations)? One could answer the question by checking the maximum value in the gauge-respective radar quantiles. Similarly, one could also find the answers to questions such as whether the samples are uniformly distributed in terms of the quantile range or just gather around the lower/higher range of the rainfall field. Without the additional information provided by radar, one would probably assign evenly distributed quantiles to the rain gauge observations as one usually does in the acquisition of an empirical CDF. See L.114-120 in the revised manuscript.

- **Question**: Here, random mixing is used as simulation method. I wonder if also another simulation method could have been used after the conditional estimation of the CDF using radar data. May be this could also briefly be discussed.

  **Answer**: Random mixing (RM) is an excellent tool that performs conditional simulation in Gaussian space, yet it is not irreplaceable. Another conditional simulation method could have been used. RM is employed in this study due to (a) the relatively high efficiency which makes mass production of realizations possible, and (b) code availability - a Python package for conditional simulation of spatial random fields using RM is available. See L.407-411 in the revised manuscript, and further Sect. 5.1 where the two cores of the proposed approach - the CDF method and RM - are discussed.

**To Remko Uijlenhoet**

**General remarks**

In the general remarks, Prof. Uijlenhoet indicates that there are two limitations, which should be stressed more clearly in the paper:

1 **Question**: It is a pure simulation study - in spite of what is suggested by the title of this paper, there are no radar or rain gauge data employed in this study.

**Answer**: The title of this paper has been modified to reveal the context of this study - "A study that implements full control over the stochastic process". An artificial experiment was carried out to test the capability of the proposed approach at estimating the true rainfall field. Due to the lack of knowledge on the true fields, we have used synthetic ones: 1000 rainfall fields were generated independently and served as the 'true' rainfall fields, from which radar and rain gauge data were derived. Unlike the commonly used verification method, e.g., leave-n-out cross-validation where one verifies the accuracy of the estimates at limited locations, in this study the accuracy of the estimates is verified more comprehensively on the overall statistics - the maximum and the mean of the simulated rainfall field. Similarly, due to the full control over the stochastic process, the sensitivity of the proposed approach towards the two factors - number of rain gauges and magnitude of random error in the radar estimates - could be analyzed. The context of this study and the motivation behind appear repeatedly in the revised manuscript: in the abstract (L.11-13), in the introduction (L.70-75), in Sect. 3 (L.223-229), in Sect. 6 (L.494-497).

2 **Question**: The study only considers the estimation of spatial rainfall fields, completely neglecting the temporal aspect of QPE, which is so important for hydrological applications.

**Answer**: The temporal aspects of QPE are outside the scope of this study, see the answer to the second question in the reply to the anonymous referee.

**Specific remarks**

1. **Question**: L.45: Hitschfeld and Bordan (1954) is a classical reference for ...

**Answer**: We have removed this reference as there is already an example of mean field bias correction schemes - Wilson (1970) in L.44-45 in the revised manuscript.

2. **Question**: L.91–97: If I understand correctly, you are forcing the intermittency in the (transformed) radar data to be the same as that of the rain gauge data. However, since both sensors (radar and gauges) have very different space-time sampling properties, ...

**Answer**: We fully agree that radar and rain gauges have different space-time sampling properties, and one should not expect the same probability of finding zeros by both sensors. Yet one should choose from the two sources of information to evaluate the intermittency. We have used radar data, as the intermittency computed from a limited number of rain gauge observations is less reliable. There is a redundancy problem. The point $(0, u_0)$ where the rainfall CDF intersects the y-axis has already been set in the previous step. Practically, we have not used the sampled zeros in both datasets in the computation of the rainfall CDF. The enforcement is not necessary in the first place. Instead they should be removed from both datasets. See Sect. 2.1 (Step 3, pg 4) in the revised manuscript.

However, the consistency at zeros is a good indicator of the mismatch between radar and rain gauge data. A significant mismatch, e.g., the collocation of a dry pixel with a 5 mm rainfall record can lead to unreliable estimates. See the discussion in L.107-109 in the revised manuscript.

3. **Question**: L.118–124: Here, 'correlation function' and 'variogram' are used interchangeably. Please choose one of the two terms as a measure of spatial dependence. Moreover, it may be true that "it can be seen that the empirical and the true variograms have very similar patterns", but that

does not mean that the truncated Gaussian fields necessarily have the appropriate (i.e. a realistic) spatial intermittency structure. Is that a problem for your approach or not? If not, why not?

**Answer**: The simulation is developed under the assumption that the correlation function of the process is stationary. Yet rather than estimating the correlation function, it is common in geostatistics to work with the variogram. It has been shown that the estimation of the variogram is more stable than the estimation of the correlation function directly. Namely, one applies the simulation using the correlation function as the measure of spatial dependence, yet the spatial dependence of the simulated product is normally examined on its variogram. Thus, we preserve the use of 'variogram' but provide an explanation why the 'variogram' appears instead of sticking with the 'correlation function', see L.141-144 in the revised manuscript.

As for the concern on the distinction between the empirical variogram (evaluated from the truncated Gaussian field) and the true variogram. It is not a problem for the approach. RM is a geostatistical simulation method whose basic element is the spatially correlated random field. Similarly, as the case in Kriging where the choice of the variogram has a limited effect on the estimates, the choice of the correlation function has a limited effect on the estimates from RM. See the brief discussion in L.138-139 in the revised manuscript. Besides, the variogram computed from the truncated Gaussian field (transformed radar data) is helpful to approach the true variogram.

4. **Question**: L.192: "George Dantzig's simplex algorithm, the BFGS method, etc." – please add literature references to these methods.

   **Answer**: We simply remove the two references due to the relatively weak association (L.210 in the revised manuscript).

5. **Question**: L.135–204: I am able to follow the general reasoning of Section 2.3, however, I am not able to grasp all the intricacies of the proposed method. It would be good if one of the referees would be familiar with the RM method, or at least if he/she would have a solid background in stochastic processes.

   **Answer**: We have made a slight improvement w.r.t. Sect. 2.3, and the code availability of RM as given in L.410-411 in the revised manuscript might be helpful.

6. **Question**: L.212–213: The authors use again the notion of the correlation function (also see Fig. 3 and caption), whereas previously they employed the variogram to express the degree of spatial dependence. For the sake of consistency, it would be good to ...

   **Answer**: see the answer to Question 3.

7. **Question**: L.217–218: "rainfall rates over an area, or the objective of this paper: short time rainfall over an area" – what do you mean exactly with "short time rainfall"? To what extent is it different from "rainfall rates"?

   **Answer**: The proposed approach is aimed at estimating spatial rainfall fields of short accumulation time: 15 min, 10 min, or even 5 min. Slight aggregation of radar data (say 2 or 3 time steps) should not change the type of distribution function remarkably. Under this assumption, it is not inappropriate to use the log-normal distribution function as the model of the CDF of the rainfall field. In L.239-240 in the revised manuscript, explicit information on the accumulation time of the short-time rainfall is given.

8. **Questions**: L.232–233: 'A random error is therefore introduced to mimic this kind of error', namely 'factors [...] that can reduce the representativeness of the radar estimates for the rainfall pattern on the ground, such as evaporation, complex terrain effects and anthropogenic influences' (L.231–232). However, many of the representativeness errors in radar rainfall estimation are not random, but systematic, e.g. range-dependent errors associated with the increasing height of the radar beam above the ground the further one goes away from the radar antenna, or the associated increase of

the radar sampling volume. How would one account for such systematic effects in the proposed simulation? How important would it be to incorporate such systematic effects?

**Answer**: The basic assumption of the method is that the radar estimates can represent the pattern of the rainfall field to a reasonable extent. Random error which can reduce the representativeness of the radar estimates is accounted for, and the quality control steps in the computation of the rainfall CDF are specifically designed to get rid of the negative effects from random error. While the degradation of the representativeness due to the systematic effects in the radar estimates, e.g., range-dependent errors associated with the increasing height of the radar beam or increasing sampling volume as one goes away from the radar antenna, is not accounted for. If the systematic effects are prevailing in the radar estimates such that the assumption is not valid, then the proposed approach is no longer applicable. In Sect. 5.3.1 (pg 21), the basic assumption of the proposed approach is discussed, and it has also been indicated that quality control to get rid of the systematic effects (e.g. range-dependent errors) in the radar estimates is necessary for a successful application of the approach.

9. **Question**: L.238–240: 'It should be notified that the introduction of random error in Step 1 differs the radar estimates from the true rainfall field in terms of the field pattern, yet the changes in the statistical properties are tiny' – see previous remark: would these 'changes' still be 'tiny' if more realistic systematic errors would be accounted for in the simulation framework, rather than random errors alone?

**Answer**: see the answer to Question 8.

10. **Question**: L.252–253: 'The choice of the two parameters - factor 0.87 and exponent 0.83 - is quite arbitrary. We have modeled a case when radar underestimates the precipitation. Surely one could model other cases.' – this approach does not appear to be based on sound assumptions ("quite arbitrary") ...

**Answer**: The purpose of applying the nonlinear transformation - $0.87\ R_r^{0.83}$ - is to mimic the error induced due to the employment of an erroneous Z-R relationship. As this relationship in changing in time, generally, there is no means to achieve the accurate one in real time.

The choice of the two parameters is indeed arbitrary, as it makes no difference for the proposed approach where the transformed radar estimates - radar quantiles - are utilized. The monotonic transformation above does not change the quantile map. Yet the choice of the two parameters matters for radar-gauge merging techniques where the radar estimates are used directly. An underestimation in the radar estimates leads to an underestimate, for example. As it is not essential for the proposed approach, we do not care too much about the choice of the two parameters. We have modeled a case when radar underestimates the precipitation, because radar data are prone to underestimate the precipitation (see the references in L.275 in the revised manuscript), and needless to say that underestimated precipitation is useless and can have negative effects for many hydrological applications. See L.272-278 in the revised manuscript.

11. **Question**: L.265: 'This gives an approximate coverage of one rain gauge for every 256, 178, and 131 km$^2$, respectively.' – why these rain gauge densities? How do they compare to rain gauge densities encountered in practice? Please motivate these choices, which (again) seem to be quite arbitrary.

**Answer**: With a simulation study, one has the opportunity to decide the layout of rain gauges, which brings both pros and cons. The advantage is that one has full control over the stochastic process which makes the sensitivity study possible. While the disadvantage is that one can hardly model the rain gauge network encountered in practice which is usually irregularly distributed with various densities. As it would be too intricate to model the varied real-world rain gauge networks, we have made things as simple as possible - square domain and uniformly distributed rain gauges. A brief discussion is given in L.283-285 in the revised manuscript. Further the limitation of this study due to the employment of synthetic data is given at the end of the revised manuscript (L.497-500).

12. **Question**: Fig. 5: These are certainly interesting simulation results, however, the generated random

fields do not really resemble actual radar-estimated rainfall fields, which often display a strong sense of directionality (anisotropy), associated with the prevailing movement direction of the rainfall field. So, how representative do you think your simulation results are for practical purposes, e.g. when applied to radar and rain gauge data from the German national weather service (DWD)? This also refers to the lacking temporal aspect of radar QPE, which was mentioned already under "general remarks".

**Answer**: Though temporal aspects of QPE are beyond the scope of this study (see the answer to the second question in the reply to the anonymous referee), the approach has potential to improve the quality of QPF (Quantitative Precipitation Forecasts), e.g., by a radar-based nowcast model. As has been proposed by Shehu and Haberlandt (2020) that the predictability loss of a nowcast model is mainly caused by the inability of radar to capture the true rainfall field and because the Lagrangian Persistence is unable to model the temporal evolution of the rainfall field. The approach could therefore be used to improve the rainfall field fed into the model, i.e., to solve the first problem. See L.415-420 in the revised manuscript.

As for the issue raised concerning directionality/anisotropy in the radar estimates (which is an interesting topic), we haven't considered the anisotropy when producing the synthetic data. Yet this property can be modeled, because the radar-related constraint is that the simulated field should resemble the radar-indicated field pattern as close as possible. If the anisotropy exists in the radar estimates, the properties can be reproduced in the simulated rainfall fields.

13. **Question**: L.306–307: 'Instead, one can obtain an infinite number of realizations for the same true rainfall field by RM' – but Kriging and other geostatistical methods can also be employed for (conditional) simulations, can't they? Or would these methods lead to the issues associated with the inaccurate representation of the marginal distribution of the generated rainfall fields, which the authors referred to in the abstract and introduction of their paper? If yes, please mention this explicitly.

With a Kriging method, one obtains a Kriged mean field which tends to underestimate the peak and overestimate the small, i.e., more middle-ranged values are present in the estimate. The Kriging method does not fix the CDF. If one evaluates the empirical CDF from the Kriged rainfall field, the departure of the empirical CDF from the true CDF can be observed. By comparison, with the proposed approach, one can obtain a bunch of simulated rainfall fields (realizations). The individual realization gives relatively accurate statistics (mean, variance, covariance). Yet practically, one can hardly obtain a single realization with accurate statistics and simultaneously with accurate locations of the rainfall peaks, as the radar-related constraint is met via optimization. An ensemble of such realizations gives a tendency towards the accurate locations of the rainfall peaks. As for the other question raised by the referee "L.295–296: 'to a certain degree' – to what degree? Please try to be as concrete as possible.", we have reworded it as 'The pattern of the true field is captured with limited accuracy.' (L.311-312 in the revised manuscript). Yet we find it difficult to provide a more concrete answer to this question.

14. **Question**: L.314: 'the mean realization is more helpful in identifying the locations of the rain cells' – OK, but this apparently goes at the expense of an unrealistic spatial variability. Can you get the locations and the variability right simultaneously with the proposed method? That would be relevant for practical applications, it seems to me.

**Answer**: In this fully controlled setup, we know exactly how the true rainfall field looks like. Yet when the true field is unknown, one possibility is that we simulate a bunch of realizations by considering both the radar and rain gauge data. The individual realization gives relatively accurate statistics (variance, covariance, etc.), and an ensemble of such realizations gives a tendency towards the accurate locations of the rainfall peaks. When feeding such an ensemble to applications such as a hydrological model, for example, one also obtains an ensemble of estimates, such that the estimation uncertainty in terms of the rainfall field propagates. See L.321-324 in the revised manuscript.

15. **Question**: Fig. 9, Table 1: Have "G25" (presumably $6 \times 6$ rain gauges), "G36" $6 \times 6$ and "G49" $7 \times 7$ been defined before? If not, please do so.

**Answer**: The abbreviations are defined in Sect. 3.3 (L.285-286) in the revised manuscript, and are repeated again in L.349.

16. **Question**: L.405: After presenting the Sensitivity Analysis (Section 4.2), which is part of the Results (Section 4), you immediately jump to the Conclusions. However, a true Discussion section, where one puts one's own results into perspective, by critically discussing assumptions and the associated limitations and by comparing obtained results with results reported elsewhere in the scientific literature, appears to be lacking. However, this is an important element of any scientific study. Therefore, I urge the authors to include such a Discussion in a revised version of this paper (unless I missed it).

    **Answer**: A discussion section is added where we discuss the two cores of the proposed approach, the scope of the approach, the basic assumption, etc. See Sect. 5 in the revised manuscript.

**Editorial remarks**

Every editorial remark has been valued and the relevant improvement according to each of them has been made.

**To Scott Sinclair**

**General comments**

Surely there are many interesting questions to be discussed concerning the applicability and capability of the proposed method. And we have added a discussion section where the scope, the assumption, etc. are discussed. And the limitation of this study, as well as an outlook are given at the end (L.494-500) of the revised manuscript.

As for the question raised by the referee "For example, does the uncertainty of estimation in figures 7c and 8 represent something more physically meaningful than the Kriging variance?", we present the answer as follows:

> The Kriging variance only reflects the relative position between the unknowns and the data points. While the estimation uncertainty by the proposed approach is more physically meaningful by considering the information and the associated uncertainty from radar data as well. The estimation uncertainty of a pixel is affected by two factors: (a) the distance of the pixel from the data points - the uncertainty from the gauge side, and (b) the expected estimate at the pixel - the uncertainty from the radar side. The tendency is clear that the closer the pixel to the neighboring rain gauge and the smaller the expected estimate at the pixel, the lower the estimation uncertainty is at the pixel. See L.328-334 in the revised manuscript.

It would indeed be of value to understand whether this technique has better performance over a range of time/space scales than other methods. And it is correct that the proposed method fixes both the marginal distribution function and the spatial correlation for each field, while for Kriging only the spatial correlation is fixed. However, this is a simulation study where synthetic data are used for verification. Regardless of the benefits, the limitation is inevitable as synthetic data can only partially represent the reality. Thus even if we present better performance of the proposed technique over a range of time/space scales than other methods, the persuasiveness is limited. In this study, the advantages of the proposed approach over the other Kriging methods is demonstrated in the experimental context, and many practical questions are to be answered; hence a further study based on realistic datasets is required. See the outlook in L.494-500 in the revised manuscript.

**Detailed and editorial comments**

- **Question**: pg 3 , line 87 - consider rewording "Uniform ... to a quantile map". The terminology is a bit confusing. Do you mean transform to a uniform distribution using a quantile map?

  **Answer**: Fixed. See L.89 in the revised manuscript.

- **Question**: pg 4, line 95 - is this quality control step justified by any reason other than practical considerations of the method?

  **Answer**: The quality control steps are specially designed to rule out the negative effects of the random error in the radar estimates. In the ideal case when the radar estimates can perfectly represent the pattern of the rainfall field, zero gauge observations and the smallest quantile $u_0$ should co-exist, and the gauge-radar pairs $(r_k, u_k)$ for $k = 1, \cdots, K$ represent K points being exactly on the rainfall CDF. Yet in practice, there exist various factors that can reduce the representativeness of the radar-indicated field pattern, which results in inconsistency at zeros, as well as the Spearman's rank correlation of the two datasets (gauge observations and the collocated radar quantiles) not being exactly 1. See the discussion in pg 4, Step 3 in the revised manuscript.

  However, the quality control steps cannot remove the systematic effects in the radar estimates (e.g. range-dependent errors). Thus quality control to get rid of the systematic effects in the radar estimates is necessary for a successful application of the proposed approach. See Sect.5.3.1 (pg 21) in the revised manuscript.

- **Question**: pg 4, figure 1 - at what spatial scale/domain size can the spatial CDF be considered valid? At some point the domain must be too large for a single CDF to represent all processes? Will the CDF be different for each time-step in a temporal simulation?

**Answer**: The rainfall CDF is changing in time. We fully agree that at some point the domain must be too large for a single CDF to represent all processes, namely, the spatial CDF can be considered valid at a limited spatial scale, and this limit should be related to factors such as rainfall regime, local climate, topography, etc. The referee has raised a very interesting question that we have not considered previously. In the experimental context of this study, we could not answer this question properly, and a further study based on realistic datasets is therefore required. See Sect. 5.3.4 (pg 22) in the revised manuscript.

- **Question**: pg 5, figure 2 - Label axes

  **Answer**: Fixed. See pg 6, Figure 2 in the revised manuscript.

- **Question**: Lines 163, 164 - how to decide to increase N?

  **Answer**: Take the case when the number of rain gauge observations is 25 as an example. We have 25 linear constraints, $K = 25$ in Eqn. (7). One should choose an initial value of $N(N > 25)$ - the number of unconditional random fields - say 50. Then solve the under-determined system defined in Eqn. (7), and find the set of weights with the least norm, $\sqrt{\sum_{i=1}^{N} \alpha_i^2}$, using, e.g., the singular value decomposition. If the norm is above a certain threshold (say 0.1), then increase N by a step of 10, for example, and solve Eqn. (7) again. Repeat the above procedure until the norm $\leq 0.1$.

- **Question**: Figure 3b - Does a different intermittency $u_0$ change the results shown in the paper in any relevant way?

  **Answer**: In this paper, the performance of the CDF method has been tested on a hydrologically interesting spatial intermittency $u_0 = 0.36$. Practically, the choice of $u_0$ has a minor influence on the performance. When $u_0 > 0.36$ (i.e. a larger dry-area-ratio), the point where the rainfall CDF intersects the y-axis moves up, and also there are more zero-samples in both the radar and rain gauge data. When $0 < u_0 < 0.36$ (i.e. a larger wet-area-ratio), the point moves down and there are fewer zero-samples. The method is problematic if $u_0 = 0$, i.e., the entire domain is wet. In that case, there is no $(0, u_0)$ and the enforcement in Step 4 (Sect. 2.1) that the CDF passes through the point $(0, u_0)$ should not be applied then. See the discussion in Sect. 5.3.3 (pg 21-22) in the revised manuscript.

- **Question**: Figure 6 - Edit caption to specify that this is for the single field example in fig 5.

  **Answer**: Fixed. See Figure 6 (pg 14) in the revised manuscript.

- **Question**: Figure 7 - the gauge layout is a uniform sampling from the field. How would a more 'realistic'/random distribution of gauges in space affect the outcome of the experiments?

  **Answer**: A uniform distribution of the rain gauges observations is not required, yet the gauge observations should not be too clustered, as it is normally considered that one has a better chance to obtain spatially representative samples from a relatively evenly distributed rain gauge network. See L.440-443 in the revised manuscript. Yet under the experimental context of this study, the answer to this question cannot be more concrete. Thus we present the limitation of this study and an outlook at the end (L.497-500) of the revised manuscript,

- **The last 3 question** are concerning the error statistics in estimating the extremes.

  **Answer**: All the results shown are about the errors in estimating the single extreme for each field, i.e., the maximum. In the revised manuscript (throughout Sect 4.2.1), we have used 'maximum' instead of 'extreme' for clarity.

  It should be clarified that when converting the simulated Gaussian field $Z$ to rainfall field $R$ using the normal-quantile transformation, i.e., Eqn. (15)

  $$R = G^{-1}(\Phi(Z))$$

the pixels that are larger than $\Phi^{-1}(0.995)$ are converted to $G^{-1}(0.995)$ to get rid of the numerical effects, where $G(\cdot)$ and $\Phi(\cdot)$ are the rainfall CDF and the CDF of the standard normal distribution, respectively. From this perspective, one could say the results shown are the extremes above a threshold quantile, though it is an extremely high threshold. If a lower threshold is used (say 0.95, 0.9, or 0.8), the bias in the results from both RM and KED should decrease. And in that case, the contrast of the results from the two methods should not be as significant as shown in Fig. 9, Line 367, and Table 1. In a nutshell, the higher the threshold quantile, the more significant the contrast is.

As for the last question "Table 1 - Also consider including the 'extremes of errors' e.g. RM shows less likely, but larger extremes in the errors. Are the extreme errors bounded to be in the same order of magnitude for both methods?", we think that the histogram of the errors provides an intuitive expression of how the errors distribute, and we have shown the most extreme cases in Fig. 9 - the scenarios that are most favorable and unfavorable for the proposed approach. Table 1 is derived from the histograms and provides supplementary information on the means and scatterings of the histograms. As for the extreme errors (the referee means the maximum absolute error if we understand correctly), it is our opinion that this statistic seems not so informative, and the two methods seem to have comparable performances, as observed from Fig. 9.